# $\mathcal{X}^2$-DFD: A framework for e$\mathcal{X}$plainable and e$\mathcal{X}$tendable Deepfake Detection

**Yize Chen**[1,2*]**, Zhiyuan Yan**[3*]**,Guangliang Cheng**[4]**, Kangran Zhao**[1,2]**, Siwei Lyu**[5]**, Baoyuan Wu**[1,2†]

[1] The Chinese University of Hong Kong, Shenzhen
[2] Shenzhen Loop Area Institute
[3] School of Electronic and Computer Engineering, Peking University, P.R. China
[4] Department of Computer Science, University of Liverpool, Liverpool, L69 7ZX, UK
[5] Department of Computer Science and Engineering,
University at Buffalo, State University of New York, Buffalo, NY, USA

## Abstract

This paper proposes $\mathcal{X}^2$-**DFD**, an **e$\mathcal{X}$plainable** and **e$\mathcal{X}$tendable** framework based on multimodal large-language models (MLLMs) for deepfake detection, consisting of three key stages (see Figure 1). The first stage, *Model Feature Assessment*, systematically evaluates the detectability of forgery-related features for the MLLM, generating a prioritized ranking of features based on their intrinsic importance to the model. The second stage, *Explainable Dataset Construction*, consists of two key modules: *Strong Feature Strengthening*, which is designed to enhance the model's existing detection and explanation capabilities by reinforcing its well-learned features, and *Weak Feature Supplementing*, which addresses gaps by integrating specific feature detectors (*e.g.*, low-level artifact analyzers) to compensate for the MLLM's limitations. The third stage, Fine-tuning and Inference, involves fine-tuning the MLLM on the constructed dataset and deploying it for final detection and explanation. By integrating these three stages, our approach enhances the MLLM's strengths while supplementing its weaknesses, ultimately improving both the detectability and explainability. Extensive experiments and ablations, followed by a comprehensive human study, validate the improved performance of our approach compared to the original MLLMs. More encouragingly, our framework is designed to be plug-and-play, allowing it to seamlessly integrate with future more advanced MLLMs and specific feature detectors, leading to continual improvement and extension to face the challenges of rapidly evolving deepfakes. Code can be found on https://github.com/chenyize111/X2DFD.

## 1 Introduction

Current generative AI technologies have enabled easy manipulation of facial identities, with many applications such as filmmaking and entertainment [52]. However, these technologies can also be misused to create *deepfakes*[2] for malicious purposes, including violating personal privacy, spreading misinformation, and eroding trust in digital media. Therefore, there is a pressing need to establish a reliable and robust system for detecting deepfakes. In recent years, numerous deepfake detection methods have been proposed [36, 45, 88, 32, 6, 60, 77, 73], with the majority focusing on addressing the generalization issue when the manipulation methods between training and testing vary. However,

---

[*]Equal contribution, [†]Corresponding author.

[2]The term "deepfake" used here refers explicitly to **face** forgery images or videos. Full (natural) image synthesis is not strictly within our scope.

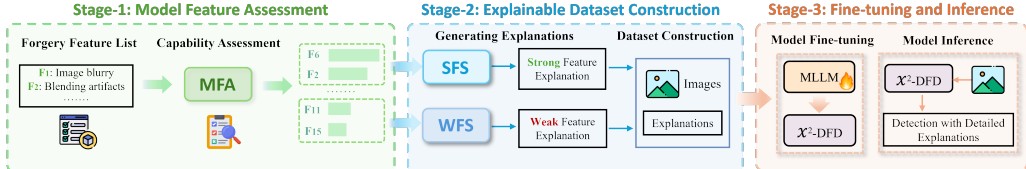

Figure 1: High-level overview of our framework, consisting of three key stages: (1) *Model Feature Assessment (MFA)* evaluates and ranks the forgery-related features (*e.g.*, blending artifacts) to generate a feature set, (2) *Strong Feature Strengthening (SFS)* enhances the model's strong features for improved detection and explanation, while *Weak Feature Supplementing (WFS)* leverages *Specific Feature Detector (SFD)* to compensate the model's weak features, and eventually resulting in an explainable dataset, and (3) The MLLM is fine-tuned using the dataset and then used for inference.

these methods typically only output a probability indicating whether a given input is AI-generated [8, 82], without providing intuitive and convincing explanations behind the prediction.

Multimodal Large Language Models (MLLMs) have shown remarkable potential in many vision tasks [70, 76, 52]. Given their strong vision-language reasoning capabilities, MLLMs offer a promising avenue for addressing the explainability gap in visual forgery detection. Recent studies [28, 59, 31, 83] have explored this direction by prompting human annotators or LLMs to describe forgery cues from multiple dimensions, which the MLLMs are then trained to detect. However, these approaches often overlook a key challenge: the **reliability** of the generated explanations. Due to MLLMs' well-documented tendency to hallucinate, especially under uncertain conditions [4], it is **crucial to ensure that the models rely on their "familiar" forgery cues with strong discrimination for detection**. Intuitively, not all forgery features are equally useful—some can be effectively leveraged for detection, while others are weakly utilized or ignored altogether.

To investigate this, we conduct a **comprehensive analysis of how well pre-trained MLLMs can utilize various forgery-related cues.** As shown in Figure 2, certain cues exhibit strong detection performance (*e.g.*, facial structures and skin tone), whereas others offer limited discriminative value (*e.g.*, blending artifacts and lighting inconsistencies). When a cue is unfamiliar or ineffective for the model, explanations based on it become unreliable. In contrast, cues that align well with the model's capabilities produce more robust and trustworthy explanations. Therefore, to ensure reliable explanations, it is essential to *explicitly* identify and promote cues that the MLLM can reliably understand and leverage.

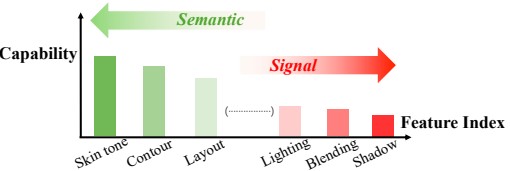

Figure 2: The diagram shows that pretrained models (*e.g.*, LLaVa) effectively distinguish real from fake content using *semantic* features (*e.g.*, Skin tone, Contour), but perform poorly with *signal* features (*e.g.*, Blending, Lighting).

Inspired by the above investigations, we propose $\mathcal{X}^2$**-DFD**, a novel framework that utilizes MLLMs for deepfake detection. The key idea of our approach is to **enhance the strengths and supplement the weaknesses** of the original MLLMs. Our framework operates through three core stages. **First**, the *Model Feature Assessment (MFA)* assesses the intrinsic capability of the original MLLMs in deepfake detection. This stage quantifies the discriminative capability of each forgery-related feature, producing a prioritized ranking based on its importance to the model. **Second**, the *Strong Feature Strengthening (SFS)* and *Weak Feature Supplementing (WFS)* reinforce strong features and compensate for weak ones, resulting in a more explainable dataset. **Third**, we use the created dataset from the second stage to fine-tune the MLLM and then use it for improved detection and explanation. This integration enables us to leverage the strengths of both MLLMs and Specific Feature Detectors (SFDs) effectively and **fuse the large and small models adaptively.** Encouragingly, the modular-based design of the proposed $\mathcal{X}^2$**-DFD** framework enables seamless integration with future MLLMs and SFDs as their capabilities evolve.

Our main contributions are threefold.

- **We systematically assess the intrinsic capabilities of MLLMs for deepfake detection**: To our knowledge, we are the first work to provide an in-depth analysis of MLLMs' inherent ability in

deepfake detection. Our findings reveal that MLLMs exhibit varying discrimination capabilities across different forgery features.

- **We enhance MLLMs' explainability by reinforcing their strong features**: Building on their strengths, we fine-tune MLLMs to generate explanations based on their most "familiar" forgery features, improving both detection accuracy and explainability.

- **We further integrate Specific Feature Detectors to supplement the model's weakness**, For forgery features where MLLMs struggle, we incorporate SFDs to complement their limitations, creating a more robust detection system.

## 2 Related Work

**Conventional Deepfake Detection**     Early detection methods typically focus on performing feature engineering to mine a manual feature such as eye blinking frequency [35], warping artifacts [36], headpose [81], and *etc*. Recent conventional deepfake detectors mainly focus on dealing with the issue of generalization [78], where the distribution of training and testing data varies. Until now, they have developed novel solutions from different directions: constructing pseudo-fake samples to capture the blending clues [36, 32, 61, 89], learning spatial-frequency anomalies [23, 45, 47, 53], focusing on the ID inconsistency clues between fake and corresponding real [17], performing disentanglement learning to learn the forgery-related features [77, 80, 22], performing reconstruction learning to learn the general forgery clues [5, 66], locating the spatial-temporal inconsistency [24, 68, 90, 79, 85], and *etc*. However, these methods can only provide real or fake predictions without providing detailed explanations. The lack of convincing and human-comprehensible explanations might confuse users about why the predictions are deemed fake.

**Deepfake Detection via Multimodal Large Language Model**     Vision and language are the two important signals for human perception, and visual-language multimodal learning has thus drawn a lot of attention in the AI community. Recently, the LLaVA series [44, 43, 42] have explored a simple and effective approach for visual-language multimodal modeling. In the field of deepfake detection, [28, 59] have investigated the potential of prompt engineering in face forgery analysis and proposed that existing MLLMs show better explainability than previous conventional deepfake detectors. In addition, [34, 21, 31] probed different MLLMs for explainable fake image detection and [34, 69] by presenting a labeled multimodal database for fine-tuning. In parallel, VIPGuard [40] explores explainable deepfake detection by leveraging identity information through an MLLM. More recently, [87] proposed using pairs of human-generated visual questions answering (VQA) to construct the fine-tuning dataset, but manually creating detailed annotations can be very costly. Addressing this limitation, [27] recently introduced an automated pipeline using GPT-4o [1] to generate VQA pairs for dataset construction and MLLM training.  However, a new critical question was then raised: *Can MLLMs (e.g., LLaVa) fully comprehend the fake clues identified by GPT-4o?* We argue that there could remain a "capability gap" between different MLLMs, particularly between "annotation generators" (GPT-4o) and "consumer models" (LLaVA). This gap exposes two unresolved challenges: (1) systematically analyzing the limitations of MLLM-based detectors in understanding all synthetic forgery clues (*e.g.*, identifying specific detection capabilities they lack) and (2) developing methods to enhance their existing strengths (*e.g.*, semantic consistency analysis) while compensating for weaknesses (*e.g.*, fine-grained artifact recognition). To our knowledge, most existing works fail to adequately address the two key challenges, leaving a critical void in building more robust and explainable deepfake detection systems.

## 3 Method

In this work, we propose a general explainable and extendable multimodal framework for deepfake detection, which consists of three key stages: (1) *Model Feature Assessment (MFA)* evaluates and ranks the forgery-related features, (2) *Strong Feature Strengthening (SFS)* enhances the model's strong features and *Weak Feature Supplementing (WFS)* leverages *Specific Feature Detector (SFD)* to compensate the model's weak features, and eventually resulting in an explainable dataset, and (3) The MLLM is fine-tuned on this dataset and then used for inference for enhanced deepfake detection and explanations. In the following content, we will introduce the technical details of these stages.

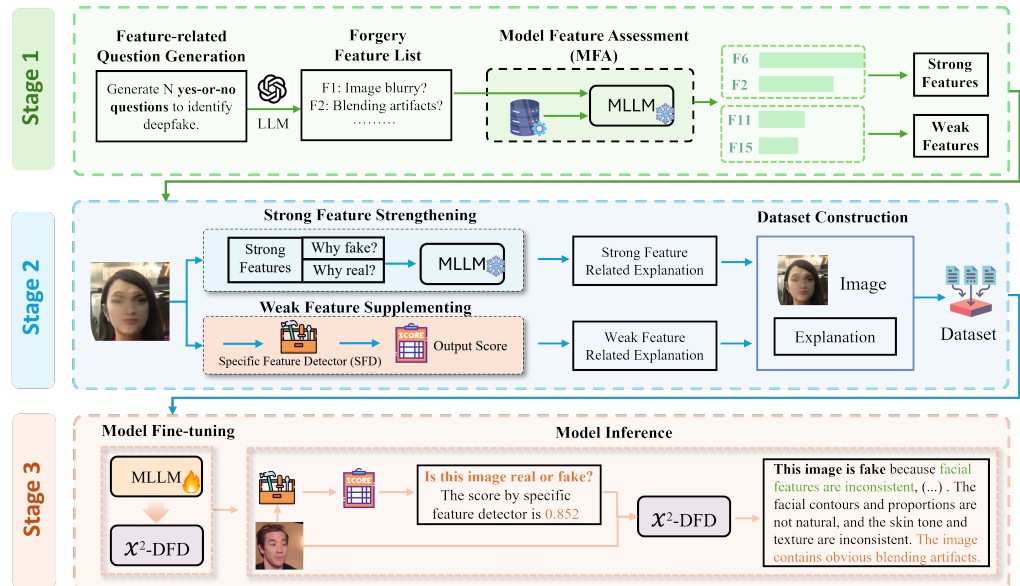

Figure 3: A comprehensive breakdown of the three-stage methodology for $\mathcal{X}^2$-**DFD**. In **Stage 1**, an automated procedure for forgery-related feature generation, evaluation, and ranking is implemented within the *MFA* (Model Feature Assessment) module. **Stage 2** incorporates the *SFS* (Strong Feature Strengthening) module, which automates the generation of explanatory annotations for a fine-tuning dataset consisting of real and fake images, leveraging strong features, alongside the *WFS* (Weak Feature Supplementing) module, which employs a specific feature detector to produce explanations for weak features. **Stage 3** entails model fine-tuning and inference, empowering the model to excel in detection performance and provide precise explanations, utilizing both its proficient strong features and less proficient weak features for improved detection and explanation.

## 3.1 Model Feature Assessment (MFA)

As depicted in the top row of Figure 3, the MFA module consists of three sequential stages: **feature-related question generation, assessment, and ranking**. Each stage plays a crucial role in identifying and prioritizing forgery-related features.

*Step 1: Feature-related Question Generation.* For each candidate forgery-related feature, a corresponding question is formulated to assess its presence in an image. Given that these features are not predefined, a Large Language Model (LLM), such as GPT-4o, is leveraged to generate a diverse set of $N_f$ questions, denoted as $F_i$. These questions are designed to probe key forgery indicators, including facial inconsistencies, unnatural color, and texture mismatches elements critical for deepfake detection.

*Step 2: Model Feature Assessment.* Each generated question is paired with an image from the assessment dataset, forming structured prompts for model inference. The Multi-Modal Large Language Model (MLLM) then responds with binary outputs (`yes` or `no`), which are aggregated into a confusion matrix to quantify the reliability of each feature. Specifically, for an image $x_j$ and question $F_i$, where $i$ represents the index of forgery feature-related question, and $j$ represents index of an image, the MLLM produces:

$$R_{i,j} = \mathcal{M}_{\text{base}}(F_i, x_j), \tag{1}$$

where $R_{i,j} \in \{\texttt{yes}, \texttt{no}\}$, representing the model's response. This step ensures that the generated questions effectively capture forgery-related discrepancies.

*Step 3: Feature Ranking.* To prioritize the most discriminative features, questions are ranked based on their *Balanced Accuracy (BA)*:

$$\text{BA}_i = \frac{1}{2}\left(\frac{\text{TP}_i}{\text{TP}_i + \text{FN}_i} + \frac{\text{TN}_i}{\text{TN}_i + \text{FP}_i}\right), \tag{2}$$

where $TP_i, TN_i, FP_i, FN_i$ denote True Positives, True Negatives, False Positives, and False Negatives, respectively. Due to potential class imbalance between real and fake samples in the dataset, we use the simple and widely adopted *Balanced Accuracy (BA)* metric. This fairly evaluates both classes, aiding effective feature ranking. The ranking identifies the most reliable forgery-related features for discrimination.

Following the automated ranking, **human verification** is further conducted to ensure the reliability of the identified fake features. This step mitigates potential biases or misinterpretations by the LLM, refining the final selection of discriminative features. Additionally, irrelevant or non-discriminative features are filtered out, with minimal instances of erroneous or unrelated outputs.

## 3.2 Strong Feature Strengthening (SFS)

The SFS module constructs datasets by leveraging the strong feature capabilities identified as high-performing by the MFA module. This process comprises two key steps.

*Step 1: Real/Fake Prompts Generation.* Leveraging the strong features from the MFA module, we generate specialized prompts to guide the MLLM's focus during the fine-tuning phase. Specifically, we first utilize GPT-4o to summarize these strong features and construct two distinct prompts: one tailored for real images ($\mathbf{P}_{real}$) and another for fake images ($\mathbf{P}_{fake}$). These prompts are formulated as: $\mathbf{P}_{real} = f(\mathbf{F}_{real}), \quad \mathbf{P}_{fake} = f(\mathbf{F}_{fake})$, where $\mathbf{F}_{real}$ and $\mathbf{F}_{fake}$ denote the sets of strong features relevant to real and fake images, respectively. Also, $f$ represents any LLMs. Here, we employ GPT-4o for implementation.

*Step 2: Fine-tuning Dataset Construction.* A fine-tuning dataset $D_{ft}$ comprising VQA-style (visual question answering) pairs, which is constructed by pairing each image with the corresponding (real or fake) prompt. Each image is annotated with the specific features it exhibits, and the standardized prompt $\mathbf{P}_{\texttt{fixed}}$ is defined as: $\mathbf{P}_{\texttt{fixed}}$ = "Is this image real or fake?" The model's response is structured to begin with a definitive statement—"This image is real/fake"—followed by an explanation based on the identified features. Formally, the final answer is represented as: $\mathbf{A}_{\texttt{final}}$ = "This image is real/fake" + $\mathbf{A}_{\texttt{real/fake}}$. Consequently, each VQA-style pair of the fine-tuning dataset $D_{ft}$ is formalized as: $\mathbf{VQA} = (\texttt{Image}, \mathbf{P}_{\texttt{fixed}}, \mathbf{A}_{\texttt{final}})$.

## 3.3 Weak Feature Supplementing (WFS)

The WFS module construct datasets by integrating specific feature detectors, which are specialized in detecting features where the MLLM shows weakness. This module follows two steps:

*Step 1: Specific Feature Detector Invocation.* For features that the MLLM identifies as weak, we deploy an specific deepfake detector (*e.g.*, a blending-based detector [41]). This specific feature detector processes the input image and generates a prediction. Note that we also employ other SFDs for implementation, and we provide an in-depth analysis for this in the Appendix G. Specifically, when utilizing a blending detector as an instance of SFD, a blending score $s$ is produced: $s = \sigma(\text{BlendDetector}(x))$, where $x$ denotes the input image, and $\sigma$ denotes the sigmoid function that transforms the logits output of the BlendDetector into the 0-1 range.

*Step 2: Integration of Specific Feature Detection Results into the Fine-tuning Dataset.*

The blending score $s$ obtained from the specific detector is incorporated into the fine-tuning dataset by appending it to the existing prompts. This is done by adding a statement such as: "And the blending feature score of content is: $\underline{s}$" Additionally, based on the score, a corresponding response aligned with the probability is included, specifically in the Fine-tuning Dataset Construction section of the SFS. This integration ensures that the MLLM benefits from both its intrinsic detection capabilities and the specialized insights provided by the SFD.

## 3.4 Model Finetune and Inference

After obtaining the constructed dataset, the following steps involve fine-tuning and inference. The stage following two steps:

*Step 1: MLLM Fine-tuning.* The initial MLLM is fine-tuned using the dataset $D_{ft}$. This process adjusts the *projector* to accurately link image artifacts with corresponding fake labels. Additionally,

Low-Rank Adaptation (LoRA) [25] is applied to selectively update a subset of the model's parameters, enhancing its focus on deepfake-specific features while preserving overall model integrity. This fine-tuning can be expressed as:

$$\mathcal{M}_{\text{base}} \xrightarrow{D_{ft}} \mathcal{M}_{\text{fine-tuned}},$$

where $\mathcal{M}_{\text{fine-tuned}}$ represents the enhanced MLLM with superior deepfake detection capabilities.

*Step 2: Integration of Specific Feature Detection into Inference Prompts.* Generally, during the inference, the SFD detector's blending score $s$ is incorporated into the MLLM's prompt-based reasoning process. The final output of the model is structured, to begin with a definitive statement: `"This image is real/fake"`, followed by reasoning based on identified visual features. Based on the blending score $s$, the model appends a descriptive statement: $\mathbf{A_{final}}$ = `"This image is real/fake"` (binary results) + $\mathbf{A_{real/fake}}$ (explanations) + `"And this image contains obvious/minimal blending artifacts"` (clues from SFD). The model acquires this response pattern through training. This approach ensures that the MLLM effectively leverages SFDs to enhance its detection performance, particularly for features where it initially demonstrated weakness.

# 4 Experiment

## 4.1 Experimental Setup

**Datasets.** We evaluate our proposed method on a diverse set of widely-used deepfake detection datasets, including the Deepfake Detection Challenge (DFDC) [15], its preview version (DFDCP) [16], DeepfakeDetection (DFD) [13], Celeb-DF-v2 (CDF-v2) [37], FaceForensics++ (FF++) [57] (c23 version for training), DFo [29], WildDeepfake (WDF) [95], FFIW [92], and the newly released DF40 dataset [75], which incorporates state-of-the-art forgery techniques such as Facedancer [56], FSGAN [50], inSwap [58], e4s [33], Simswap [7], and Uniface [91]. In line with the standard deepfake benchmark [78], we use the c23 version of FF++ for training and other datasets for testing in the main table. Additionally, we evaluated a broader range of facial forgery types using the DiFF [9] dataset, a comprehensive collection of diffusion-generated facial images, which allowed us to test our method on a wider spectrum of forgery techniques.

**Evaluation Metrics.** We assess the performance of our model in terms of both detection performance and explanation quality. **For detection**, we adopt the Area Under the Curve (AUC) as the primary metric to evaluate the model's ability to distinguish real from fake content across entire datasets, reporting both frame-level and video-level AUC scores. Additional metrics, including Accuracy (Acc.), Equal Error Rate (EER), and Average Precision (AP), are also provided for a comprehensive analysis. **For explanation**, we follow [87] by using human-annotated data to measure text similarity between model-generated explanations and human-labeled ground truth, employing standard metrics such as *BLEU* [51], *CIDEr* [65], *ROUGE-L* [38], *METEOR* [14], and *SPICE* [2]. Beyond text similarity, we engage human evaluators and GPT-4o to assess the quality of explanations regarding forgery content, following prior studies [72, 21]. Evaluators rate the explanations on a scale from 0 (very poor) to 5 (excellent), ensuring a robust qualitative evaluation.

**Implementation Details.** We initialize our model with the LLaVA-base weights and fine-tune the LLaVA model [44] using its official implementation codebase. For the specific feature detectors (SFD), we adopt a blending-based approach as proposed in [41]. Training is performed on a single NVIDIA 4090 GPU for 3 epochs, with a learning rate of $2 \times 10^{-5}$ in two layer mlp projector and $2 \times 10^{-4}$ in others, a rank of 16, and an alpha value set conventionally to twice the rank at 32. We use a batch size of 4, a gradient accumulation step of 1, and a warmup ratio of 0.03 to stabilize training.

## 4.2 Generalizability Evaluation

Following the common settings of previous works [73, 10], we first compare our method with 33 SOTA detectors (including both conventional and multimodal-based detectors) via **cross-dataset evaluations** (see Table 1.) The results of other compared baselines are mainly cited from their original papers. Our approach excels across both frame-level and video-level evaluations, maintaining superior results when compared to other methods. The table clearly highlights our method's capability to generalize and consistently achieve higher detection performance at the frame level and video level, respectively.

Table 1: Cross-dataset evaluations with 33 existing detectors. The top two results are highlighted, with the best in **bold** and the second-best underlined. '*' indicates our reproductions based on the pre-trained checkpoints released by the authors, and '†' refers to the MLLM-based detectors, which can also output explanations. ‡: The LAA-Net is trained on the high-quality FF++ (raw) data, whereas our method is trained on the compressed (c23) version.

| | | Frame-Level AUC | | | | | | Video-Level AUC | | | | |
|---|---|---|---|---|---|---|---|---|---|---|---|---|
| **Method** | **CDF** | **DFDCP** | **DFDC** | **DFD** | **Avg.** | **Method** | **CDF** | **DFDCP** | **DFDC** | **DFD** | **Avg.** | |
| Xception [12] | 73.7 | 73.7 | 70.8 | 81.6 | 75.0 | Xception [12] | 81.6 | 74.2 | 73.2 | 89.6 | 79.7 | |
| FWA [36] | 66.8 | 63.7 | 61.3 | 74.0 | 66.5 | PCL+I2G [89] | 90.0 | 74.4 | 67.5 | – | – | |
| Efficient-b4 [64] | 74.9 | 72.8 | 69.6 | 81.5 | 74.7 | LipForensics [24] | 82.4 | – | 73.5 | – | – | |
| Face X-ray [32] | 67.9 | 69.4 | 63.3 | 76.7 | 69.3 | FTCN [90] | 86.9 | 74.0 | 71.0 | 94.4 | 81.6 | |
| F3-Net [53] | 77.0 | 77.2 | 72.8 | 82.3 | 77.3 | ViT-B (CLIP) [54] | 88.4 | 82.5 | 76.1 | 90.0 | 84.3 | |
| SPSL [45] | 76.5 | 74.1 | 70.1 | 81.2 | 75.5 | CORE [49] | 80.9 | 72.0 | 72.1 | 88.2 | 78.3 | |
| SRM [47] | 75.5 | 74.1 | 70.0 | 81.2 | 75.2 | SBI* [60] | 90.6 | 87.7 | 75.2 | 88.2 | 85.4 | |
| ViT-B (IN21k) [19] | 75.0 | 75.6 | 73.4 | 86.4 | 77.6 | UIA-ViT [94] | 82.4 | 75.8 | – | 94.7 | – | |
| ViT-B (CLIP) [54] | 81.7 | 80.2 | 73.5 | 86.6 | 80.5 | SLADD* [6] | 79.7 | – | 77.2 | – | – | |
| RECCE [5] | 73.2 | 74.2 | 71.3 | 81.8 | 75.1 | DCL [63] | 88.2 | 76.9 | 75.0 | 92.1 | 83.1 | |
| IID [26] | 83.8 | 81.2 | – | – | – | SeeABLE [30] | 87.3 | 86.3 | 75.9 | – | – | |
| ICT [18] | 85.7 | – | – | 84.1 | – | CFM [46] | 89.7 | 80.2 | 70.6 | 95.2 | 83.9 | |
| LSDA [73] | 83.0 | 81.5 | 73.6 | 88.0 | 81.5 | UCF [77] | 83.7 | 74.2 | 77.0 | 86.7 | 80.4 | |
| VLFFD† [62] | 83.2 | 83.2 | – | **94.8** | – | NACO [85] | 89.5 | – | 76.7 | – | – | |
| FFAA† [27] | – | – | 74.0 | 92.0 | – | AltFreezing [68] | 89.5 | – | – | – | – | |
| RepDFD† [39] | 80.0 | **90.6** | 77.3 | – | – | TALL-Swin [71] | 90.8 | – | 76.8 | – | – | |
| MFCLIP † [86] | 83.5 | 86.1 | – | – | – | StyleDFD [11] | 89.0 | – | – | 96.1 | – | |
| KFD-VLM † [84] | 89.9 | 86.7 | – | 92.3 | – | LAA-Net ‡ [48] | 95.4 | 86.9 | - | **98.4** | - | |
| $\mathcal{X}^2$-**DFD (7B)** | 90.4 | 87.3 | **83.7** | 92.3 | 88.4 | $\mathcal{X}^2$-**DFD (7B)** | 95.4 | 89.3 | **86.0** | 95.8 | 91.6 | |
| $\mathcal{X}^2$-**DFD (13B)** | **91.3** | 90.3 | 83.4 | 92.5 | **89.4** | $\mathcal{X}^2$-**DFD (13B)** | 95.7 | 91.0 | 85.7 | 96.1 | **92.1** | |

## 4.3 Explainability Evaluation

**Annotated Explainability Evaluation.** We assess the performance of our model using the DD-VQA [87] test dataset, which incorporates human-annotated data from FF++ [57]. The evaluation employs a suite of metrics, including *BLEU* [51], *CIDEr* [65], *ROUGE-L* [38], *METEOR* [14], and *SPICE* [2], to quantify the alignment between our model's responses and human-annotated ground truth. The MLLMs assessed for explanation quality include LLaVA [43], Llama3.2V [20], Qwen2VL [67] and GPT4o [1]. The models use the same prompt to generate explainable outputs. To ensure a fair comparison, particularly given GPT-4o's tendency to refuse responses with the same prompt, we adopt a prompting strategy from [28]. This leads GPT-4o to generate shorter responses, resulting in lower scores. The evaluation results are summarized in Table 2 *(Annotated Explainability Evaluation)*. Due to the DD-VQA only annotating the artifact in some specific parts (*e.g.*, *nose, eyes*), GPT-4o and human experts are required to evaluate both annotated and unannotated scenarios, ensuring a thorough assessment of model explainability across different scenarios.

Table 2: Explainability evaluation across annotated and unannotated settings, comparing five scores for annotated explainability, alongside human and GPT-4o evaluations (scored 0-5) for unannotated explainability. The best result per metric is highlighted in **bold**.

| Model | Annotated Explainability | | | | | | Unannotated Explainability | | |
|---|---|---|---|---|---|---|---|---|---|
| | **BLEU** | **CIDEr** | **ROUGE-L** | **METEOR** | **SPICE** | **Avg.** | **Human-Eval** | **GPT4-Eval** | **Avg.** |
| LLaVA-7B [44] | 0.183 | 0.021 | 0.139 | 0.110 | 0.085 | 0.108 | 2.368 | 1.542 | 1.955 |
| Llama3.2-Vision [20] | 0.131 | 0.009 | 0.131 | 0.081 | 0.116 | 0.093 | 2.265 | 1.667 | 1.966 |
| Qwen2.5-VL-7B [67] | 0.140 | 0.012 | 0.143 | 0.081 | 0.150 | 0.105 | 2.034 | 1.383 | 1.709 |
| GPT-4o [1] | 0.123 | 0.011 | 0.082 | 0.051 | 0.072 | 0.068 | 2.559 | 2.055 | 2.307 |
| Ours | **0.203** | **0.027** | **0.155** | **0.148** | **0.155** | **0.138** | **3.572** | **2.668** | **3.120** |

**Unannotated Explainability Evaluation.** To evaluate unannotated explainability, we build on insights from prior work [72, 21, 93] and utilize both human evaluators and GPT-4o to assess model performance across three key dimensions: (1) detection ability, (2) reasonableness of explanations, and (3) level of detail. Each dimension is scored on a scale from 0 to 5. The evaluation results are

summarized in Table 2 *(Unannotated Explainability Evaluation)*. The results demonstrate that our model achieves strong performance in both human-eval and GPT-eval. Additional details include the experiment of the setting of human study, Graphical User Interface (GUI) of human study, and the evaluation prompt of GPT4o can be found in Appendix.

## 4.4 Computational Complexity Evaluation

We performed a simple evaluation of the time complexity. Compared to other pretrained MLLMs, the model does not require much additional inference time in the same inference framework, as seen in Table 3, making the time acceptable. However, when compared to conventional Non-MLLM methods, the model, despite gaining interpretability and a richer feature space, requires more inference time. Nevertheless, we believe that as MLLM inference frameworks advance, along with pruning and quantization lightweight methods, MLLMs will benefit from these improvements when applied to Deepfake detection.

Table 3: Comparison of average inference time per image for different models.

| Model | Seconds |
|---|---|
| Non-MLLMs - Xception | ~0.03 |
| Non-MLLMs - CDFA | ~0.05 |
| Llama-3.2-11B | ~3.3 |
| Qwen2.5 VL-Instruct 7B | ~1.6 |
| LLaVa-7B | ~1.2 |
| Ours (X2DFD w/ two SFDs) | ~1.3 |

## 5 Ablation Study and Analysis

Here, we address several **key research questions** through ablation studies and in-depth analysis.

*Question 1: Why is fine-tuning MLLMs with their strong features more effective than using all?*

To enhance the reasoning and detection capabilities of MLLMs, we introduce the Strong Feature Strengthening (SFS) module. In this module, we focus on selectively amplifying the most discriminative forgery-related features—referred to as strong features. These strong features are identified through Model Feature Attribution (MFA), which ranks features based on their importance scores across different modalities and samples. Fine-tuning on these "strong features" is more effective because it focuses the learning process on highly discriminative, reliable forgery-related cues while avoiding the disruptive noise introduced by weak features.

This design leads us to a critical question: *Are these strong features truly more effective for improving model performance than using the full set of features?* To answer this, we compare two strategies: (1) enhancing all features in the LLM-generated feature list and (2) enhancing only the strong features selected via MFA. As shown in Table 4, the latter consistently outperforms the former across all datasets.

Table 4: Comparison of model detection performance and interpretation between using all features and strong features, both enhanced by the Strong Feature Strengthening (SFS) module. The evaluation metrics include AUC and GPT-4o evaluation. Additionally, results for the Top-K feature selection strategy and the No Feature Explanation are provided for comparison.

| Model Configuration | CDF | Uniface | HPS-Diff | Avg | GPT4o Eval |
|---|---|---|---|---|---|
| No Feature Explanation | 80.3 | 81.9 | 84.7 | 82.3 | – |
| X2DFD (Top-K=25) | 83.0 | 84.3 | 87.1 | 84.8 | 2.91 |
| **X2DFD (Top-K=50)** | **83.2** | **84.5** | **88.7** | **85.5** | **3.02** |
| X2DFD (Top-K=75) | 81.9 | 83.2 | 84.6 | 83.2 | 2.77 |
| X2DFD (Top-K=100) | 79.0 | 82.3 | 83.6 | 81.6 | 2.63 |

These results confirm that selectively strengthening the most discriminative features not only improves the model's performance but also yields more reliable model explanations.

*Question 2: How to ensure SFS module works when it should, and stays silent when it shouldn't?*

To further improve model generalization and interpretability, we extend our framework with two new components: the Specific Feature Detection (SFD) module and the Weak Feature Supplementing (WFS) module. While the earlier SFS module focuses on enhancing strong, highly discriminative features, it may overlook subtle patterns critical for certain forgery types. To address this, WFS is designed to teach the model how to lever-

Table 5: Effect of excluding supplementary features during training (WFS) and including them at inference (SFD infer) on model performance.

| Varient | CDF | DFD | DFDC | DFDCP | Avg. |
|---|---|---|---|---|---|
| WFS ✗ SFD infer ✗ | 83.2 | 91.4 | 79.2 | 82.0 | 84.0 |
| WFS ✗ SFD infer ✓ | 81.7 | 90.6 | 79.1 | 81.3 | 83.2 |
| WFS ✓ SFD infer ✓ | **90.4** | **92.3** | **83.7** | **87.3** | **88.4** |

age weak features provided by SFD—features that are otherwise hard for MLLMs to interpret directly.

To investigate whether this combination yields synergistic benefits, we compare three variants: (1) baseline without WFS and SFD at inference, (2) enabling SFD only during inference (without WFS), and (3) enabling both WFS and SFD. As shown in Table 5, the model achieves the best performance when WFS is present, demonstrating that **SFD's weak signals become more useful once the model has learned how to utilize them through WFS**. Without WFS, simply adding SFD at inference may not help—and can even lead to degradation—indicating that "1+1" only becomes greater than 2 when weak features are integrated structurally during training.

Beyond synergy, it is **crucial to ensure that the introduction of SFD does not interfere in scenarios where its cues are irrelevant**. For example, blending-based detectors may provide limited value on datasets like SRI, which contain no blending traces.

Table 6: Comparison of AUC performance for models trained on FF++ alone versus FF++ with SRI, evaluated on other datasets.

| Train Data | CDF | DFDCP | DFDC | DFD | Uniface | Fsgan | Inswap | Simswap | Avg. |
|---|---|---|---|---|---|---|---|---|---|
| FF++ ✓ SRI ✗ | 90.4 | 87.3 | 83.7 | 92.3 | 85.5 | **91.1** | **81.2** | 85.1 | 87.1 |
| FF++ ✓ SRI ✓ | **91.5** | **89.3** | **83.9** | **92.7** | **87.4** | 89.9 | 81.0 | **86.1** | **87.7** |

We test this by introducing SRI as a training set and comparing performance. As shown in Table 6, the model not only maintains its effectiveness but even improves, suggesting that **when SFD signals are weak or absent, the model naturally downplays them**. This demonstrates that our design is adaptive—SFD helps when it can, and steps aside when it should.

Overall, our framework achieves both **synergistic improvement and non-intrusive integration**: *WFS enables the model to benefit from weak features without forcing reliance, and SFD contributes only when its signals are relevant.*

*Question 3: How can we generate the most suitable set of N forgery-related questions in MFA?*

The Model Feature Assessment (MFA) module evaluates the model's discriminative ability by asking it to answer a curated set of N forgery-related questions. *A key challenge here is: how to generate the most suitable questions that effectively probe the model's understanding of diverse forgery cues.* To explore this, we compare different question-generation strategies: **(1)** human-written features based on expert knowledge [87], **(2)** features automatically generated by large language models (LLMs), including Claude3.5-Sonnet [3] and GPT-4o [1].

As shown in Table 7, questions generated by LLMs slightly outperform those crafted by humans in terms of detection performance across multiple datasets. This suggests that LLMs can capture a broader and potentially more nuanced range of forgery-related features, possibly including cues overlooked by human experts.

Table 7: Comparison between LLMs and human annotators for generating N forgery-related questions for MFA.

| Variant | CDF | DFDCP | DFDC | DFD | Uniface | Fsgan | Simswap | Avg |
|---|---|---|---|---|---|---|---|---|
| Human Writing [87] | 89.1 | 89.7 | 83.6 | 92.5 | 82.3 | 89.1 | 87.0 | 87.6 |
| Claude3.5-Sonnet [3] | 90.1 | 88.5 | 83.5 | 93.0 | 84.9 | 90.0 | 85.6 | 87.9 |
| GPT4o [1] | 90.3 | 89.7 | 83.5 | 92.5 | 85.2 | 89.9 | 84.9 | 87.8 |

However, LLM-generated questions are not always ideal—they may sometimes be generic, redundant, or irrelevant (e.g., mistakenly treating "Photoshop traces" as core deepfake features). On the other hand, although human-designed features may be narrower in scope, they offer higher precision and domain relevance, leading to robust results. To balance these strengths and weaknesses, we adopt a **hybrid strategy**: **(1)** Use an LLM to generate a diverse pool of forgery-related questions. **(2)** Rank them by relevance scores. **(3)** Apply human verification to filter out irrelevant or low-quality questions.

*Question 4: Can framework benefit from extending multiple SFDs?*

With the continuous development of generative technologies, diffusion-based generative methods and techniques have been emerging rapidly. To test the scalability of our framework, we extended it to another SFD, AlignedForensics [55], and incorporated a new training dataset, consisting of fake faces generated through diffusion techniques from 1000 external images, which form a subset of the face-swapping data in DiFF [9]. We then tested not only the face-swapping scope, which was previously the focus of this work (e.g., CDF and Simswap), but also expanded the evaluation to include face editing, image-to-image editing, and text-to-image generation using the DiFF dataset, which also focuses on faces.

As shown in Table 8, we found that the X2DFD framework not only effectively extended the CDFA's blending-based SFD but also enhanced the performance when extending both the CDFA and AlignedForensics (diffusion-based SFD). This extension allowed the framework to achieve strong results across various forgery types and methods.

Table 8: AUC Performance Evaluation for SFD Integration X2DFD-Sig (CDFA only) vs. X2DFD-Mul (CDFA & AlignedForensics)

| Model | CDF | Simswap | Diff-FE | Diff-I2I | Diff-T2I |
|---|---|---|---|---|---|
| CDFA | 87.9 | 76.1 | 74.6 | 81.7 | 87.3 |
| X2DFD-Sig | 90.4 | 85.1 | 82.1 | 81.7 | 88.3 |
| X2DFD-Mul | 90.3 | 88.5 | 92.1 | 88.6 | 92.2 |

## 6 Conclusion

In this paper, we propose $\mathcal{X}^2$-DFD, a **unified multimodal framework for *explainable* and *extendable* deepfake detection**. For the first time, we systematically evaluate the intrinsic capabilities of the pre-trained MLLMs, revealing their varying effectiveness across different forgery-related features. Inspired by this, we implement a targeted fine-tuning strategy, which has largely improved the explainability of the MLLMs, specifically capitalizing on their strengths. Furthermore, by integrating specific feature detectors (SFD), we design an adaptive fusion module to combine the complementary advantages of both MLLMs and conventional detectors for improved detection.

**Limitations and Future Work.**   While our framework demonstrates strong performance in detecting identity-specific facial forgeries, it has certain limitations. **First**, multimodal large language models (MLLMs) operate in a much larger parameter space, which leads to a richer visual feature space related to Deepfake detection. However, this comes at the cost of slower inference speeds. Future advancements in the field could help accelerate inference, benefiting from the rapid development of the domain. **Second**, our current implementation focuses solely on static image detection. However, real-world applications increasingly involve multimodal forgeries across video and audio streams. Extending our method to handle videos and audio-visual deepfakes is a critical next step for building a comprehensive and practical detection system. **Third**, As forgery technologies advance and realistic deepfakes exhibit fewer detectable artifacts that can be captured by natural language descriptions, our MLLM-based method may become less effective in interpreting semantic artifacts, thereby requiring the SFD module to play a more crucial role in the framework.

**Broader Impacts.**   This research advances machine learning with a new framework to detect and explain Deepfake images, effectively identifying deepfakes and reducing misuse of generative models for significant societal benefit. However, it risks being used to improve deepfake realism. To counter this, following previous works [74, 75] implement access controls. We will urge researchers to minimize harms while maximizing the positive impact of this work.

**Ethics & Reproducibility statements.**   All facial images used are from publicly available datasets with proper citations, ensuring no violation of personal privacy. And the human study has received the *IRB approval*.

**Content Structure of the Appendix.**   Due to page constraints, we include additional analyses and experiments in the Appendix. Specifically, the Appendix contains the following sections: Overview of Appendix, Experiment Setting Details, Additional Experimental Results, Human Study and GPT4 Evaluation, Additional Analysis of Model Feature Assessment, Additional Analysis of Strong Feature Strengthening, Additional Analysis of Weak Feature Supplementing, Additional Analysis of Ablation Study, Sample Showing. **For further details, please refer to the Appendix.**

## Acknowledgments

Baoyuan Wu was supported by Guangdong Basic and Applied Basic Research Foundation (No. 2024B1515020095), Guangdong Provincial Program (No. 2023TQ07A352), Shenzhen Science and Technology Program (No. JCYJ20240813113608011), and Longgang District Key Laboratory of Intelligent Digital Economy Security.

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

# A Overview

Due to space constraints, we have included additional important content in the supplementary materials. Below is a brief outline of the supplementary content to facilitate readers easily locate the relevant sections:

- Appendix B: Experiment Setting Details
  - Appendix B.1: Details of Datasets
  - Appendix B.2: Implementation Details
- Appendix C: Additional Experimental Results
  - Appendix C.1: Results of Robustness Against Unseen Perturbations
  - Appendix C.2: Results of Training and In-domain Testing on FF++
  - Appendix C.3: Results of Cross-manipulation Evaluation on DF40
  - Appendix C.4: Results of Experiments on Different LLMs/MLLMs
  - Appendix C.5: Results of Experiments on GenAI Images
- Appendix D: Human Study and GPT4 Evaluation
  - Appendix D.1: Human Study Details
  - Appendix D.2: GPT-4 Evaluation Details
- Appendix E: Additional Analysis of Model Feature Assessment
  - Appendix E.1: Evaluation Setup of MFA
  - Appendix E.2: Evaluation of the Overall Detection Performance Details
  - Appendix E.3: Deeply Investigation of Individual Feature's Discrimination
- Appendix F: Additional Analysis of Strong Feature Strengthening
- Appendix G: Additional Analysis of Weak Feature Supplementing
- Appendix H: Additional Analysis of Ablation Study
- Appendix I: Sample Showing

# B Experiment Setting Details

## B.1 Details of Datasets

**VQA Datasets Construction Details**   In this part, we detail the data construction process for our deepfake detection framework, $\mathcal{X}^2$-DFD, which integrates multiple stages: Multi-Feature Analysis (MFA), Strong Feature Selection (SFS), Weak Feature Selection (WFS), and Model Inference (MI). The process, illustrated in Figure 4, leverages multimodal large language models (MLLMs) and specific feature detectors (SFD) to construct a robust dataset for training and fine-tuning.

The data construction pipeline begins with the MFA stage, where an MLLM generates yes-or-no questions (*e.g.*, "Is the image blurry?" or "Are there blending artifacts?") to identify deepfake characteristics in a given dataset. These questions are evaluated by the MLLM, producing a feature score list that ranks features by their relevance to deepfake detection. In the SFS stage, the top-K features are selected to generate real and fake prompts, such as providing reasoning for why an image might be deepfake or real. These prompts are paired with the deepfake dataset to construct a fine-tuning dataset, where the MLLM answers questions like "Is this image real or fake?" to generate labeled data. The WFS stage further refines this dataset by incorporating an External Dedicated Detector (SFD), which assigns confidence scores (*e.g.*, SFD score = 0.901 for fake samples) to filter out unreliable samples, ensuring the dataset's quality for fine-tuning. Finally, in the MI stage, the fine-tuned MLLM, now strengthened as $\mathcal{X}^2$-DFD, performs inference on new samples, producing accurate deepfake detection results (*e.g.*, "This image is fake" with detailed reasoning about blending artifacts and omitted features). This multi-stage pipeline ensures that the constructed dataset is both comprehensive and reliable, enabling the $\mathcal{X}^2$-DFD model to achieve robust performance across various deepfake detection scenarios.

**Training and Testing Image Datasets**   We trained our model using the FF++ dataset [57]. For preprocessing and cropping, we adopted the methods from DeepfakeBench [78]. We utilized 8 frames per video for training and 32 frames per video for testing.

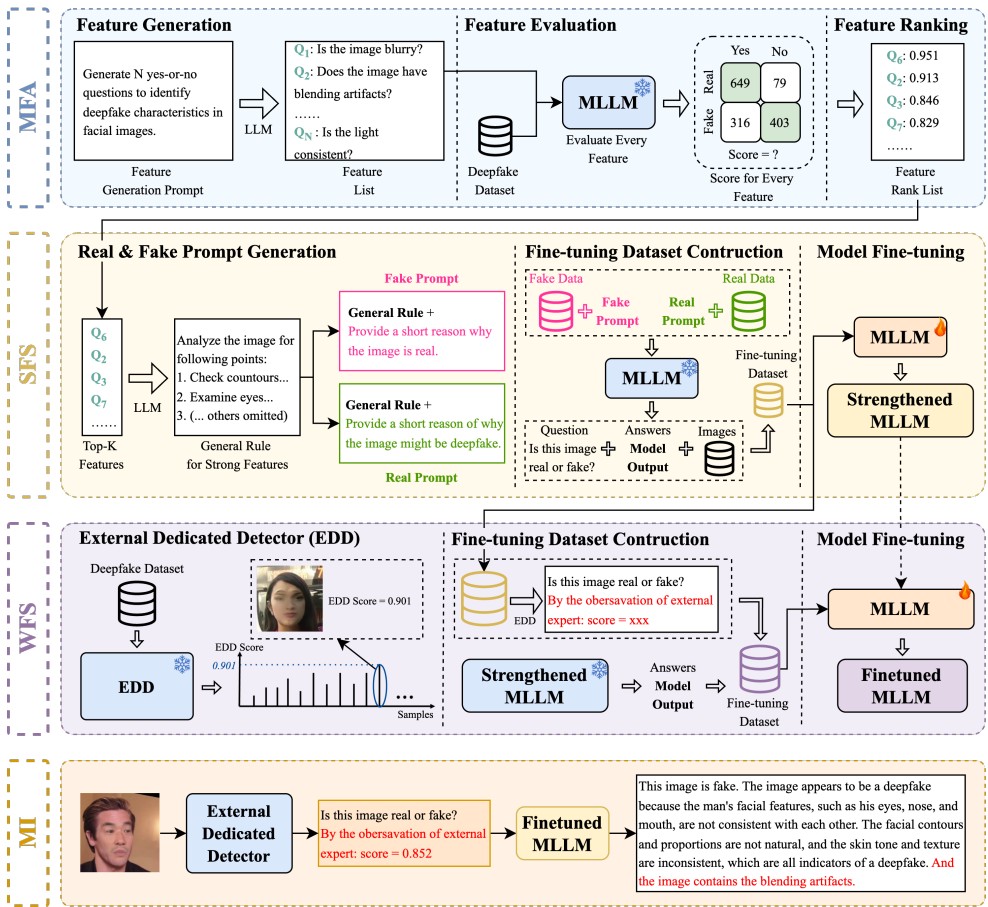

Figure 4: Overview of the data construction pipeline for the $\mathcal{X}^2$-DFD framework. The pipeline consists of four stages: (1) Multi-Feature Analysis (MFA) for generating yes-or-no questions to identify deepfake characteristics, resulting in a feature list; (2) Strong Feature Selection (SFS) for creating real and fake prompts based on top-K features, followed by dataset construction using MLLM-generated answers; (3) Weak Feature Selection (WFS) for fine-tuning the dataset with SFD scores, filtering samples based on reliability; and (4) Model Inference (MI) for final evaluation.

## B.2 Implementation Details

The data splits and preprocessing follow the DeepfakeBench [78]. We train with a learning rate of $2 \times 10^{-5}$ in two layer mlp projector and $2 \times 10^{-4}$ in others which is borrow from the official implementation of LLava [43], a rank of 16, and an alpha value set conventionally to twice the rank at 32. We training for three epochs, for each epoch on NVIDIA 4090 (Driver Version: 535.247.01; CUDA Version: 12.2), AMD 32-Corecost for 4 hours by training on FF++ [57], each video we take 8 frames for training. AUC is calculated by directly obtaining the token probabilities. Previous AUC calculations for large models mostly relied on averaging methods, such as in [28], but this approach is not very accurate because: (1) multiple samplings are needed to approximate the true probability distribution, and (2) large models inherently perform inference with a default temperature, which itself involves sampling over probabilities. Averaging over multiple samples effectively results in a second layer of sampling, making the evaluation less accurate. Therefore, in this paper, we calculate AUC by directly obtaining token probabilities.

# C Additional Experimental Results

## C.1 Results of Robustness Against Unseen Perturbations

To evaluate the robustness of our model to random perturbations, we follow the methodology outlined in previous studies [24, 90], which examines four types of degradation: Gaussian blur, block-wise distortion, contrast changes, and JPEG compression. Each perturbation is applied at five different levels to assess the model's performance under varying degrees of distortion.

To highlight the advantages of our approach over conventional detectors like FWA [36], SBI [60], and X-ray [32], we conducted multiple evaluations. As illustrated in Figure 5, which shows the video-level AUC results for these unseen perturbations using a model trained on FF++ c23, our method consistently demonstrates superior robustness compared to other RGB-based methods.

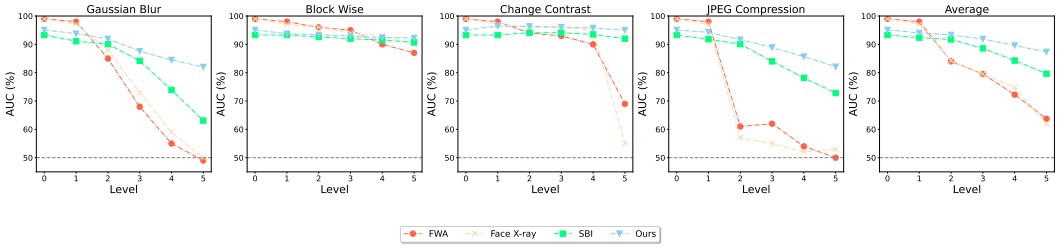

Figure 5: Robustness evaluation. We adopt four types of degradation for examining the robustness of our model: Gaussian blur, block-wise distortion, contrast changes, and JPEG compression. Our model shows superior robustness over other compared models.

## C.2 Results of Training and In-domain Testing on FF++

In our manuscript, we mainly focus on the cross-domain evaluation to assess the generalization performance of different models. Here, we conduct the in-domain evaluation on the FF++ dataset and compare our approach with the other four SOTA methods: FWA, Face X-ray, SRM, and CDFA. Following DeepfakeBench [78], we train all models on FF++ (c23) and test them on FF++ (c23), FF++ (c40), FF-DF, FF-F2F, FF-FS, and FF-NT. As shown in Table 9, the in-domain results demonstrate that our framework achieves the best performance, outperforming all other methods.

Table 9: In-domain results in the FF++ dataset (AUC)

| Detector | FF++$c23$ | FF++$c40$ | FF-DF | FF-F2F | FF-FS | FF-NT | AVG |
|---|---|---|---|---|---|---|---|
| FWA [36] | 87.7 | 73.6 | 92.1 | 90.0 | 88.4 | 81.2 | 85.5 |
| Face X-ray [32] | 95.9 | 79.3 | 97.9 | 98.7 | 98.7 | 92.9 | 93.9 |
| SRM [47] | 95.8 | 81.1 | 97.3 | 97.0 | 97.4 | 93.0 | 93.6 |
| CDFA [41] | 90.2 | 69.0 | 99.9 | 86.9 | 93.3 | 80.7 | 90.2 |
| Ours | 96.6 | 82.6 | 99.9 | 97.2 | 98.1 | 91.0 | 94.2 |

## C.3 Results of Cross-manipulation Evaluation on DF40

Evaluating our model's performance on cross-manipulation tasks helps assess whether it can handle previously unseen fake types. We use the recently released DF40 dataset [75] for evaluation. Our method generally outperforms other models on average, particularly the e4s, Inswap, and SimSwap methods (see Table 10). This shows that our method effectively learns more generalizable features for detection, even against the latest techniques.

## C.4 Results of Experiments on Different LLMs/MLLMs

We conducted experiments using various models, including GPT4o [1], Claude3.5-Sonnet [3], and LLaVa [44], to evaluate the adaptability and robustness of our framework.

Table 10: Cross-manipulation evaluations within the FF++ domain (frame-level AUC only). We leverage the DF40 dataset [75] and select six representative face-swapping methods generated within the FF++ domain, keeping the data domain unchanged. The top two results are highlighted, with the best result in **bold** and the second-best underlined.

| Method | Venues | uniface | e4s | facedancer | fsgan | inswap | simswap | Avg. |
|--------|--------|---------|-----|------------|-------|--------|---------|------|
| RECCE [5] | CVPR 2022 | 84.2 | 65.2 | 78.3 | 88.4 | 79.5 | 73.0 | 78.1 |
| SBI [60] | CVPR 2022 | 64.4 | 69.0 | 66.7 | 87.9 | 63.3 | 56.8 | 68.0 |
| CORE [49] | CVPRW 2022 | 81.7 | 63.4 | 71.7 | 91.1 | 79.4 | 69.3 | 76.1 |
| IID [26] | CVPR 2023 | 79.5 | 71.0 | 79.0 | 86.4 | 74.4 | 64.0 | 75.7 |
| UCF [77] | ICCV 2023 | 78.7 | 69.2 | 80.0 | 88.1 | 76.8 | 64.9 | 77.5 |
| LSDA [73] | CVPR 2024 | 85.4 | 68.4 | 75.9 | 83.2 | 81.0 | 72.7 | 77.8 |
| CDFA [41] | ECCV 2024 | 76.5 | 67.4 | 75.4 | 84.8 | 72.0 | 76.1 | 75.9 |
| ProDet [10] | NIPS 2024 | 84.5 | 71.0 | 73.6 | 86.5 | 78.8 | 77.8 | 78.7 |
| **Ours** | – | **85.5** | **90.3** | **82.6** | **91.1** | **81.2** | **85.1** | **85.9** |

**Different LLMs to Generate Questions in MFA.** In the MFA stage, we employed different LLMs, such as GPT4o and Claude 3.5-Sonnet, to generate forgery-related questions and test the adaptability of our framework. The results, shown in *GPT4o + LLaVa-7B* and *Claude 3.5-Sonnet + LLaVa-7B*, demonstrate consistent performance regardless of the LLM used. Questions from Claude 3.5-Sonnet were also effective (see Table 21 and 22).

Table 11: Experiments on different LLMs/MLLMs were conducted to evaluate their performance under various conditions. The evaluation metric used for these experiments is the Area Under the Curve (AUC)

| Variant | CDF | DFDCP | DFDC | DFD | Uniface | e4s | Facedancer | FSGAN | Inswap | Simswap | Avg |
|---------|-----|-------|------|-----|---------|-----|------------|-------|--------|---------|-----|
| Human Writing + LLaVa-7B [43] | 89.1 | 89.7 | 83.5 | 92.5 | 83.3 | 86.1 | 82.5 | 89.1 | 78.7 | 87.0 | 86.2 |
| GPT4o[1] + LLaVa-7B [43] | 90.3 | 89.7 | 83.5 | 92.5 | 85.2 | 91.2 | 83.8 | 89.9 | 78.5 | 84.9 | 87.0 |
| GPT4o[1] + Qwen2VL-7B [67] | 89.8 | 90.3 | 82.9 | 93.3 | 84.9 | 91.2 | 81.6 | 90.1 | 79.9 | 84.0 | 86.8 |
| Claude3.5-Sonnet[3] + LLaVa-7B [43] | 90.1 | 88.5 | 83.5 | 93.0 | 84.9 | 90.6 | 83.8 | 90.0 | 80.0 | 85.6 | 87.0 |
| GPT4o[1] + LLaVa-13B [43] | 91.3 | 90.3 | 83.4 | 92.5 | 86.0 | 92.5 | 84.5 | 91.0 | 80.6 | 85.4 | 87.8 |

**Different MLLMs for Fine-tuning in SFS and WFS.** In the SFS and WFS stages, we investigate the impact of using different sizes of MLLMs with the same architecture during fine-tuning. For instance, we compare *GPT4o + LLaVa-7B* and *GPT4o + LLaVa-13B*. The results indicate that as the size of the model increases, the performance of the framework in Table 11 improves proportionally, benefiting from the enhanced capabilities of the larger MLLMs. Additionally, we tested *Qwen2VL-7B*, which yielded performance comparable to *LLaVa-7B*.

**Comparison of Human-generated and Model-generated questions.**

We also compared the effectiveness of human-written questions with those generated by advanced LLMs, such as GPT4o and Claude 3.5-Sonnet, in the MFA stage. The results on Table 11, exemplified by *Human Writing + LLaVa-7B* (Avg AUC 86.2) versus *GPT4o + LLaVa-7B* (Avg AUC 87.0) and *Claude 3.5-Sonnet + LLaVa-7B* (Avg AUC 87.0), reveal that human-generated questions perform comparably to model-generated ones. Notably, questions produced by LLMs often exhibit greater competitiveness, leveraging their ability to generate diverse and nuanced forgery-related prompts, which further enhances the framework's adaptability and performance.

**Summary of Findings.** These experiments collectively highlight the robustness of our framework. It is not dependent on specific LLMs or MLLMs, making it adaptable to a wide range of models. Furthermore, as the performance and size of the underlying models improve, our framework effectively leverages these advancements to achieve enhanced results.

## C.5  Results of Experiments on GenAI Images

In addition to face swap forgery, we further explored the effectiveness of our framework on GenAI Image. Please refer to the **supplementary material** for more details and examples.

# D    Human Study and GPT4 Evaluation

## D.1    Human Study Details

The human study was designed to evaluate the explainability of our deepfake detection model by involving human evaluators in assessing three key dimensions: detection ability, reasonableness of explanations, and level of detail. Below are the key details of the experimental setup:

**Participant Recruitment.**    We recruited 15 well-educated participants and provided them with a detailed guideline to ensure a clear understanding of the experimental task. Participants were aged between 20 and 40.

**Task Description.**    Participants were presented with a set of 100 samples, each consisting of a deepfake image, the model's detection output ($\mathcal{X}^2$-**DFD**), and an associated explanation generated by a Multimodal Large Language Model (MLLM). They were tasked with scoring the model's performance across three dimensions on a scale from 0 to 5:

- *Detection Ability*: How accurately does the model identify the deepfake? (0 = completely incorrect, 5 = perfectly accurate)
- *Reasonableness of Explanations*: How logical and understandable is the explanation? (0 = completely unreasonable, 5 = highly reasonable)
- *Level of Detail*: How detailed and specific is the explanation? (0 = no detail, 5 = very detailed)

**Study Procedure.**    The study was conducted using a custom-built GUI. Participants completed an initial 10-minute training session to familiarize themselves with the task and scoring criteria. Each participant evaluated random 50 samples, ensuring diverse coverage of the dataset. The study took approximately 20 minutes per participant, and participants volunteered their time without compensation.

**Dataset.**    All samples were sourced from the testing datasets of DD-VQA. Each sample included both the raw image and the model-generated explanation, with a focus on unannotated regions.

**Evaluation Metrics.**    Scores for each dimension were averaged across all participants. Additionally, participants provided a final overall score for each sample on a scale from 0 to 5, categorized as follows: 0 (very poor), 1 (poor), 2 (fair), 3 (good), 4 (very good), and 5 (excellent).

**GUI of Human Study.**    he human study was conducted using a custom-designed Graphical User Interface (GUI) to facilitate efficient and user-friendly evaluation. The GUI was developed using Python with the flask library and hosted on a server. Below are the details of the GUI design: The GUI consisted of three main panels (see Figures 6 to 9 for screenshots):

- *Image Panel*: Displayed the deepfake image on the left side of the screen, with zoom and pan functionality for detailed inspection.
- *Explanation Panel*: Presented the model's detection output ($\mathcal{X}^2$-**DFD**) and other tested MLLLM's explanations on the right side.
- *Scoring Panel*: Provided sliders for each answer allowing participants to select a score from 0 to 5.

**User Interaction.**    Participants could navigate between samples using "Next" and "Previous" buttons. A progress bar indicated the number of samples completed. The GUI automatically saved scores to a database after each submission.

## D.2    GPT-4 Evaluation Details

In addition to the human study, we conducted evaluations using GPT-4o to comprehensively assess the unannotated explainability of our deepfake detection model. This subsection provides the

## Answer Evaluation

**Background:** Deepfakes are synthetic media, typically videos or images, created using advanced artificial intelligence techniques like deep learning to manipulate or generate realistic-looking content, often by superimposing one person's face onto another's, raising concerns about misinformation and privacy.

**Ground Truth: fake**

**Description:** The image is fake because it is a manipulated or digitally altered photo of a woman with a nose that is too large for her face. This distortion is evident in the facial features and the overall appearance of the image. The alteration creates an unnatural and artificial look, which is not representative of a real person. This manipulation is typically done to create a specific visual effect or to exaggerate certain features, but in this case, it leads to an unnatural and unconvincing representation of the woman's face.

**Please evaluate the model's response based on three key dimensions:**

1. **Detection Accuracy:** Did the model answer the same as the Ground truth?
2. **Reasonableness of Explanations:** How plausible and well-founded are the model's explanations?
3. **Level of Detail:** Does the model provide sufficient and appropriate detail in its analysis?

**Based on the above criteria, please provide your final assessment of the model's response.**

**Final Score:**

| 0 Very Poor | 1 Poor | 2 Fair | 3 Good | 4 Very Good | 5 Excellent |

Previous   Next

Figure 6: Human evaluation

## Answer Evaluation

**Background:** Deepfakes are synthetic media, typically videos or images, created using advanced artificial intelligence techniques like deep learning to manipulate or generate realistic-looking content, often by superimposing one person's face onto another's, raising concerns about misinformation and privacy.

**Ground Truth: real**

**Description:** The image is real because it shows a man with a beard and mustache, wearing a white shirt, and making a surprised or shocked expression. The man's facial features, such as his eyes, nose, and mouth, appear to be in their natural state, and there are no noticeable glitches or artifacts in the image. The man's expression and the overall composition of the image suggest that it is a genuine, real-life moment captured in the photo.

**Please evaluate the model's response based on three key dimensions:**

1. **Detection Accuracy:** Did the model answer the same as the Ground truth?
2. **Reasonableness of Explanations:** How plausible and well-founded are the model's explanations?
3. **Level of Detail:** Does the model provide sufficient and appropriate detail in its analysis?

**Based on the above criteria, please provide your final assessment of the model's response.**

**Final Score:**

| 0 Very Poor | 1 Poor | 2 Fair | 3 Good | 4 Very Good | 5 Excellent |

Previous   Next

Figure 7: Human evaluation

implementation details of the automated evaluation process using GPT-4o, including the code structure, prompt formatting, and result processing.

The evaluation process was implemented in Python, leveraging the OpenAI API to interact with GPT-4o. The script performs the following key steps:

**Data Loading.** The script loads a JSON file containing metadata for the evaluation samples, including image IDs, ground truth labels (*e.g.*, "real" or "fake"), and model-generated descriptions.

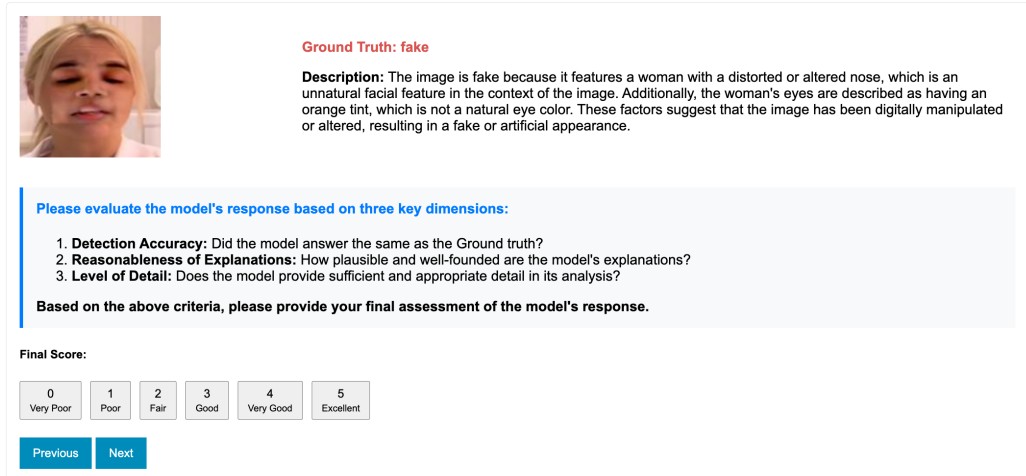

Figure 8: Human evaluation

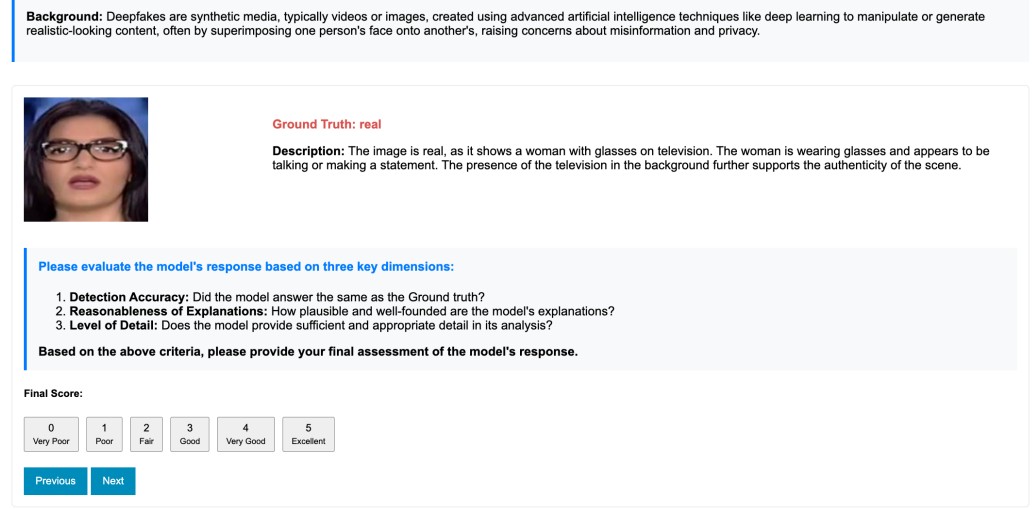

Figure 9: Human evaluation

**Prompt Formatting.** For each sample, the script formats an evaluation prompt that includes the ground truth label and the model's description (the blue guideline as shown in Figure 6). The prompt instructs GPT-4o to evaluate the model's response across three dimensions—detection accuracy, reasonableness of explanations, and level of detail—and to provide scores on a 0–5 scale (0 = Very Poor, 5 = Excellent). Additionally, a final overall score is requested.

**Image Encoding.** The script encodes each image into Base64 format to enable GPT-4o to process both the textual prompt and the visual content. Images are sourced from the directory

**Result Processing and Storage.** The script processes all 100 samples, storing the results in a list. Each result entry includes the image ID, image path, ground truth, prompt, and GPT-4o's response. The final output JSON file provides a comprehensive record of GPT-4o's evaluations, which are later aggregated and compared with human scores.

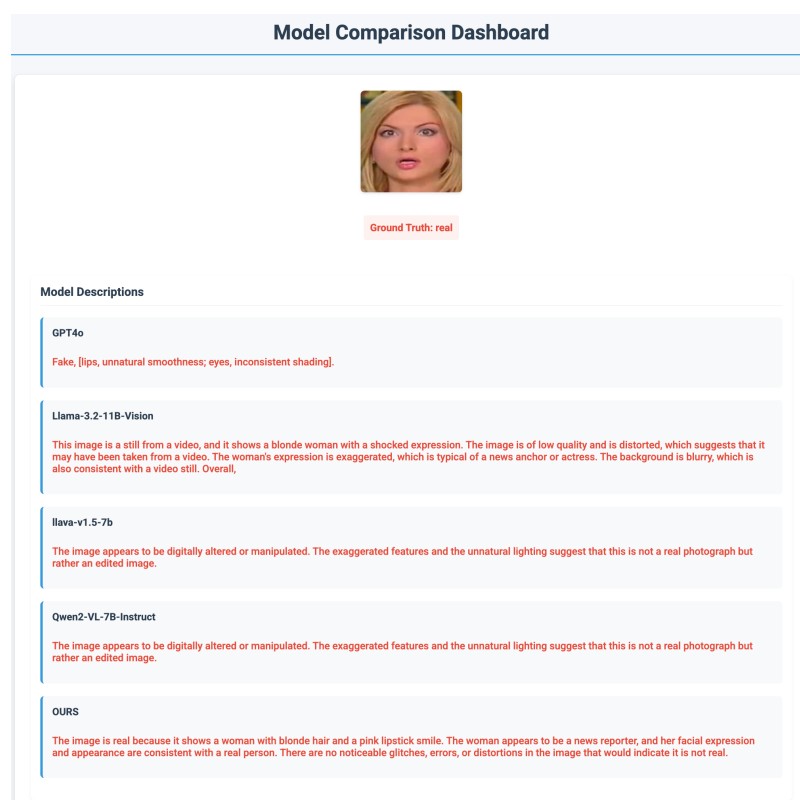

Figure 10: Answer from different model

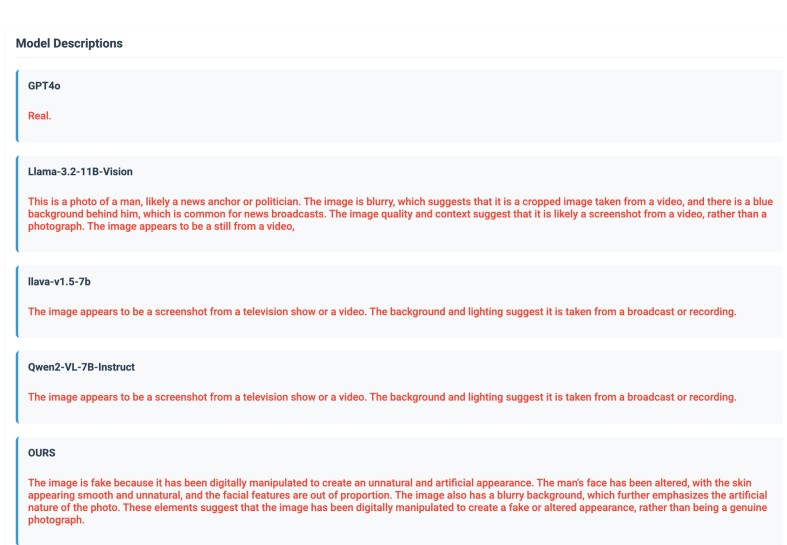

Figure 11: Answer from different model

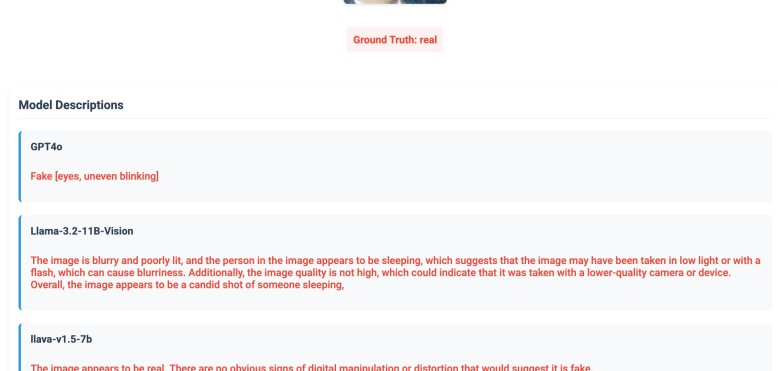

Ground Truth: real

**Model Descriptions**

**GPT4o**

Fake [eyes, uneven blinking]

**Llama-3.2-11B-Vision**

The image is blurry and poorly lit, and the person in the image appears to be sleeping, which suggests that the image may have been taken in low light or with a flash, which can cause blurriness. Additionally, the image quality is not high, which could indicate that it was taken with a lower-quality camera or device. Overall, the image appears to be a candid shot of someone sleeping,

**llava-v1.5-7b**

The image appears to be real. There are no obvious signs of digital manipulation or distortion that would suggest it is fake.

**Qwen2-VL-7B-Instruct**

The image appears to be real. There are no obvious signs of digital manipulation or distortion that would suggest it is fake.

**OURS**

The image is real because it shows a man with a nose, mouth, and other facial features that are clearly visible and not distorted or altered. The man is wearing a jacket and has a close-up shot of his face, which indicates that the image is not a manipulated or digitally altered photo. The presence of a nose, mouth, and other facial features in the image suggests that it is a genuine representation of the man's appearance, rather than a manipulated or altered image.

Figure 12: Answer from different model

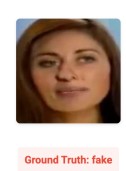

Ground Truth: fake

**Model Descriptions**

**GPT4o**

Fake, [face, unnatural texture and blending issues around the skin and eyes].

**Llama-3.2-11B-Vision**

It's a bit blurry and low quality, but I can see that the woman's face is distorted and the eyes are not properly proportioned to the rest of her face. The image appears to have been manipulated, and I'm not sure why. The woman's mouth appears to be too wide and the eyes appear to be too far apart. The image appears to be a fake image. It may have been created for a specific purpose,

**llava-v1.5-7b**

The image appears to be real. It shows a person with long hair, but the context and background are not clear enough to determine the authenticity of the image.

**Qwen2-VL-7B-Instruct**

The image appears to be real. It shows a person with long hair, but the context and background are not clear enough to determine the authenticity of the image.

**OURS**

The image is real because it shows a woman with a nose that is not perfectly aligned with the rest of her face. This imperfection is noticeable and adds to the authenticity of the image. The woman's facial features are not perfectly symmetrical, and there are no signs of distortion or manipulation that would indicate it is a fake or altered image.

Figure 13: Answer from different model

```
Project Summary:

With the rapid development of Deepfake technology, the generation of fake
images and videos has become increasingly realistic, posing significant
threats to society and individual privacy. Despite the availability of
various deep learning models for detecting Deepfake content, there are still
notable shortcomings in terms of explainability and transparency. These
differences directly impact human trust and understanding during the
detection process.

This project aims to evaluate the alignment between our model's
```

```
explainability in Deepfake detection and human intuition through a human
evaluation study. We will compare our model with current mainstream Deepfake
detection models to examine the intuitiveness, accuracy, and user trust in
the explanations provided. The study will involve participants evaluating
the detection results and explanations from different models, assessing the
effectiveness of each model in real-world applications.
```

Figure 14: Human study material part1

```
Project Significance:

This research will reveal the strengths and weaknesses of different Deepfake
detection models in terms of explainability, especially in alignment with
human intuition. By comparing our model with other mainstream models, we
hope to demonstrate that our model is not only competitive in detection
accuracy but also achieves higher trust and satisfaction in terms of
explainability. The project outcomes will offer new insights and directions
for the future design and application of Deepfake detection models,
promoting the development of more transparent and reliable detection
technologies.
```
```
Potential Side Effects, Hazards, and Emergency Plans:

Side Effects: During the project, participants may experience confusion
or reduced trust in real images due to exposure to a large amount of Deepfake
content, leading to difficulty in distinguishing between real and fake
images.
Emergency Plan: We will inform participants that there are currently
various effective Deepfake detection solutions, including the model we are
developing, which can effectively counter Deepfake attacks to some extent.
Through education and explanation, we will help participants understand the
progress of the technology and the available protective measures, enhancing
their trust in real images.
```

Figure 15: Human study material part2

```
Potential Ethical Issues and Countermeasures (including Informed Consent,
Privacy Protection, Physical Harm, and Benefit Distribution):

Informed Consent: Participants need to fully understand the research
content, purpose, and potential impact. We will ensure that all participants
voluntarily sign informed consent forms. Countermeasure: Provide detailed
research explanations and Q&A sessions to ensure participants fully
understand and agree to participate.

Privacy Protection: All participant information and data involved in the
research will be kept strictly confidential and will not be used for any
purposes other than the research. Countermeasure: Implement data
encryption and anonymization to ensure participant privacy is not
```
```
compromised.

Physical Harm: Although this study mainly involves psychological and
cognitive assessments, we will minimize potential psychological impacts on
participants and avoid any form of psychological stress or harm.
Countermeasure: Conduct real-time monitoring during the study to ensure
participant mental health, and provide participants with the right to
withdraw from the study at any time.

Benefit Distribution: Ensure that participants are not unfairly treated
during the study and that the outcomes of the research do not
disproportionately benefit or disadvantage any individual or group.
Countermeasure: Fair and reasonable distribution of research outcomes,
ensuring openness and transparency. The research results will primarily
focus on academic and social contributions, not for personal or commercial
gain.
```

Figure 16: Human study material part3

# E  Additional Analysis of Model Feature Assessment

## E.1  Evaluation Setup of MFA

**Model.**  We choose the mainstream MLLM, $i.e.$, LLaVA [44] as the implementation instance of the pre-trained MLLMs. Additionally, we choose one classical detector, Xception [12], as a baseline model for comparison.

**Dataset.**  We evaluate the models on several widely-used deepfake datasets, including the Deepfake Detection Challenge (DFDC) [15], the preview version of DFDC (DFDCP) [16], DeepfakeDetection (DFD) [13], Celeb-DF-v2 (CDF-v2) [37], DF40 dataset [75], which incorporates state-of-the-art forgery techniques such as Facedancer [56], Fsgan [50], Inswap [58], e4s [33], Simswap [7], and Uniface [91]. providing a comprehensive foundation for evaluating overall detection performance.

**Evaluation Metrics.**  We use the Area Under the Curve (AUC) as the primary evaluation metric, enabling us to assess the model's ability to distinguish between real and fake images across the whole dataset. In this section, we use the frame-level AUC for evaluation. For individual feature discrimination, we focus on forgery-related features such as "Is the face layout unnatural?" with responses of either "yes" or "no." The proportions of "yes" and "no" answers for real and fake images are calculated as follows, with the ranking score $S^{(q)}$ defined based on the balanced accuracy of the responses:

$$S^{(q)} = \frac{1}{2} \left( \frac{Y_{\text{real}}^{(q)}}{Y_{\text{real}}^{(q)} + N_{\text{real}}^{(q)}} + \frac{N_{\text{fake}}^{(q)}}{Y_{\text{fake}}^{(q)} + N_{\text{fake}}^{(q)}} \right). \tag{3}$$

Here, $Y_{\text{real}}^{(q)}$ and $Y_{\text{fake}}^{(q)}$ denote the number of "yes" answers, while $N_{\text{real}}^{(q)}$ and $N_{\text{fake}}^{(q)}$ represent the number of "no" answers for real and fake, respectively. This formulation ensures that both true positive and true negative rates are considered, providing a balanced measure of feature discrimination.

## E.2  Evaluation of the Overall Detection Performance

The comparison between LLaVA [44] and Xception [12] highlights a notable performance gap. Results in Figure 17 (left) indicate that the average AUC for LLaVA is 63.7%, while Xception achieves 75.8%, showing a notable gap of 12.1% points. This suggests that, while the LLaVA has certain zero-shot capabilities in other tasks such as (general) image classification, it is still not as strong as the traditional detector in detecting deepfakes.

However, LLaVA shows strong detection abilities in specific methods (eg., e4s), sometimes even surpassing Xception (see Figure 17 (left)). This motivates us to further investigate its intrinsic detection capabilities, and understand the model's "strengths and weaknesses" in deepfake detection. Below, we provide a detailed investigation of the discrimination of each forgery-related feature.

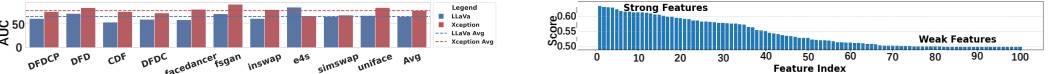

Figure 17: (Left) AUC comparison between (zero-shot) LLaVA (blue) and Xception (red) for deepfake detection across different datasets; (Right) Balance accuracy score for individual feature discrimination, with Strong features in the top-left corner and Weak features in the bottom-right corner based on discrimination scores. Full questions/features are provided in the Table.

## E.3  Deeply Investigation of Individual Feature's Discrimination

**Step 1: Question Generation.** For each candidate forgery-related feature, we formulate a corresponding interrogative statement. For instance, the feature "*blurry*" is transformed into the question "*Is the image blurry?*". Recognizing that the candidate features are not pre-specified by developers, we employ a Large Language Model (LLM), $i.e.$, GPT-4o, to automatically generate a comprehensive list of $F_i$ questions. These questions target key forgery indicators, including but not limited to lighting anomalies, unnatural facial expressions, and mismatched textures, which are critical for identifying deepfakes.

**Step 2: Question Evaluation.** Each generated question is paired with an image from the assessment dataset to form a prompt for constructing the fine-tuning dataset. The model responds with a binary output ("yes" or "no") based on its interpretation of the image in relation to the question. These responses are aggregated into a confusion matrix for each question, thereby quantifying the detection capability of the associated forgery-related features. Mathematically, for each question $F_i$ and image $x_j$, the MLLM produces:

$$R_{i,j} = \mathcal{M}_{\text{base}}(F_i, x_j), \tag{4}$$

where $R_{i,j} \in \{\text{yes}, \text{no}\}$, representing the model's response for each image-question pair.

**Step 3: Question Ranking.** According to the accuracy of all candidate questions, we obtain a descending ranking of questions, *i.e.*, the ranking of forgery-related features. This ranking allows us to quantify how well each feature contributes to distinguishing between real and fake images. Specifically, the accuracy of each question is computed by evaluating the proportion of correct responses across the dataset. Specifically, for each question $Q_i^k$, We calculate the true positive rate (TPR) and true negative rate (TNR), then take their average to obtain the Balanced Accuracy, as follows:

$$\text{Balanced Accuracy}_i = \frac{1}{2}\left(\frac{\text{TP}_i}{\text{TP}_i + \text{FN}_i} + \frac{\text{TN}_i}{\text{TN}_i + \text{FP}_i}\right), \tag{5}$$

where: $\text{TP}_i$ denotes True Positives for question $F_i$, $\text{TN}_i$ the True Negatives for question $F_i$, $\text{FP}_i$ the False Positives for question $F_i$, and $\text{FN}_i$ the False Negatives for question $F_i$.

Subsequently, questions are ranked in descending order based on their balance accuracy scores, thereby prioritizing forgery features that effectively discriminate between real and fake images.

**Strong Features.** Strong features typically involve *semantic-level facial structural or appearance anomalies*. As shown in the strong feature section of Fig. 17 (right), which primarily includes facial irregularities such as unusual facial layouts (eg., *Rank 9, 11, 17*) or distorted facial features (eg., *Rank 3, 4, 14*), eg., the nose, eyes, or mouth. Since the pre-trained MLLM is good at extracting and utilizing these features for detection, it can provide a more reliable and accurate explanation.

**Weak Features.** Weak features typically involve *fine-grained, low-level textures*, such as blending anomalies. As shown in Fig. 17 (right), these weak features are primarily subtle details related to texture, reflection, shadow, and blending. Examples of texture issues include rough or overly smooth surfaces (eg., *Rank 68, 77, 83*). Furthermore, inconsistencies in lighting and shadows (eg., *Rank 85, 86, 90, 96*) and blending artifacts on the face (eg., *Rank 54, 84, 88*) are also prominent. Since these signal-level anomalies are challenging for pre-trained MLLMs to detect, the pre-trained MLLM is likely to struggle in reliably distinguishing between real and manipulated content when relying on these weak features for detection and explanation.

**Feature Score.** The scores for different forgery-related features are presented, where Table 19 highlights the top 50 strong features, and Table 20 shows the 50 weakest features based on their scores.

### Does the model know these features are related to deepfake?

We used a series of questions to query the model, applying simple prompt augmentation with the feature-related questions mentioned above. A "yes" indicates the model knows these features are related to deepfake detection, while a "no" indicates the model does not. Detailed results are shown in Table 17 and Table 18.

## F   Additional analysis of Strong Feature Strengthening

Following the Model Feature Assessment (MFA) module, we observed significant performance improvements in forgery detection after applying the Strong Feature Strengthening (SFS) module, The generalization performance is improved by 20% AUC (see Table 14) on average compared to the pretrain model.

we conducted a detailed analysis by re-evaluating the model using the Model Feature Assessment (MFA) module to compare the discriminative capability of forgery-related features before and after

applying the SFS module. As illustrated in Figure 18, over 60% of the features exhibited enhanced discriminative power post-SFS, with particularly notable improvements in strong features.

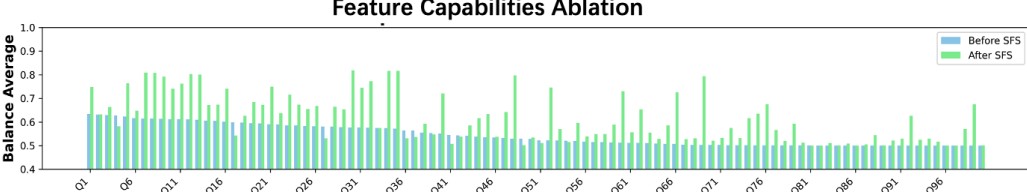

Figure 18: Comparison of feature capability before and after SFS. After adding the external detector to supplement the MLLM, the model's feature capabilities (almost all) can be further improved.

## G  Additional Analysis of Weak Feature Supplementing

In addition to the used blending model [41], we also try other instances to implement the SFDs in the WFS module of our framework, each targeting specific types of artifacts, where SRI focusing on the generative artifacts by deep nets, F3-Net [53] focusing on the frequency-level anomalies, and SBI and CFDA focusing on the blending boundaries. Based on these empirical attempts, We summarize the **general criteria** under which conditions the selected SFD instance can be used in our framework. Specifically, **the integrated SFD instance should meet the following criteria:**

- **Criteria-1**: Each SFD instance should focus on only one type of feature that is positively correlated with fake;
- **Criteria-2**: The score given by the SFD instance can accurately reflect the characteristics of the corresponding feature;
- **Criteria-3**: The data distribution of this feature in the dataset is relatively uniform.

Below, we show a detailed illustration of using other SFD instances for implementation one by one.

**AIGC Expert Integration.** We first consider implementing an AIGC expert to learn the deep generative artifacts. For implementation, we introduce the *SRI model*, based on self-reconstruction images generated by Simswap [7] and train on the *Xception model*, designed to capture self-reconstruction generative features. However, from Figure 19, integrating this model into our framework results in only a minor performance improvement of 0.3%. Further analysis reveals a negative correlation between the model's features and fake labels in the training set (do NOT meet the **Criteria-1**), indicating that these artifacts are poorly represented in the training data. Consequently, the model struggles to leverage the expert-provided features effectively, offering limited benefits over not using the expert model.

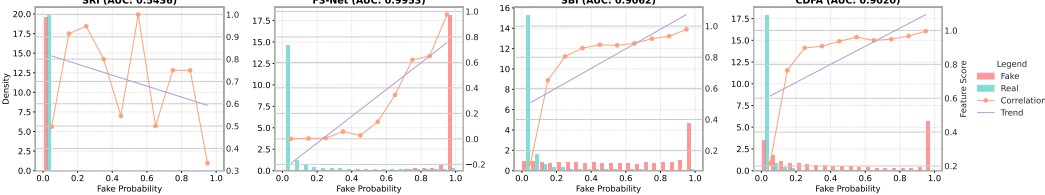

Figure 19: The probability distributions of different expert models on the FF++ training dataset. From left to right, the models are SRI, F3-Net, SBI, and CDFA, corresponding to experts in capturing self-reconstruction, frequency anomalies, and self-blending artifacts, respectively. The blending here directly uses the trained weights.

**Frequency Expert Integration.** We then integrate a frequency-based model *F3-Net* and train it on the FF++ dataset [57] to capture frequency anomalies. However, from Figure 19, the overall model's performance is identical to that of the expert, with no improvement. Although the expert features are positively correlated with fake labels, the frequency-based scores are overfitted to the training set and do not accurately reflect the true feature quantity, with only near-1 (1 for fake) and near-0 (0 for real) predictions (Not satisfy the **Criteria-2**) This leads to a *shortcut*, where the model relies solely on the

expert's output without learning from the feature information, thus limiting the extendability of the integrated model.

Table 12: Comparison of methods across datasets with values rounded to two decimal places, where the evaluation metric is AUC, The "Diff" column shows the difference from the mllms average.

| Variant | CDF | DFDCP | DFDC | DFD | Uniface | e4s | Facedancer | FSGAN | Inswap | Simswap | Avg | Diff |
|---------|-----|-------|------|-----|---------|-----|------------|-------|--------|---------|-----|------|
| MLLM | 83.3 | 82.0 | 79.2 | 91.4 | 84.5 | 94.1 | 79.9 | 88.0 | 77.2 | 83.3 | 84.3 | 0.0 |
| SRI | 42.9 | 49.3 | 52.9 | 50.9 | 97.3 | 65.7 | 71.3 | 80.1 | 80.5 | 99.9 | 69.1 | -15.2 |
| SRI+MLLM | 83.2 | 82.5 | 77.6 | 88.8 | 85.6 | **95.8** | 81.9 | 88.5 | 77.9 | 84.6 | 84.6 | +0.3 |
| F3Net | 77.0 | 77.2 | 72.8 | 82.3 | 87.5 | 71.6 | 75.4 | 89.2 | 83.9 | 77.2 | 79.4 | -4.9 |
| F3Net+MLLM | 76.8 | 77.8 | 73.1 | 83.3 | 88.4 | 75.5 | 76.6 | 89.8 | 84.6 | 78.4 | 80.4 | -3.9 |
| SBI | 82.1 | 82.3 | 70.5 | 85.5 | 83.4 | 76.8 | 68.5 | 83.2 | 77.4 | 87.7 | 79.7 | -4.6 |
| SBI+MLLM | 88.6 | 85.5 | 75.6 | 90.8 | **88.7** | 93.7 | 77.6 | 88.2 | 81.6 | **91.1** | 86.2 | +1.9 |
| CDFA | 87.9 | 86.6 | 83.5 | 90.9 | 76.5 | 67.4 | 75.4 | 84.8 | 72.0 | 76.1 | 80.1 | -4.2 |
| CDFA+MLLM | **90.3** | **89.7** | **83.5** | **92.5** | 85.2 | 91.2 | **83.8** | **89.9** | 78.5 | 84.9 | **87.0** | **+2.7** |

**SBI and CFDA Models Integration.** We also integrate another blending-based expert model, *SBI* [60], which specializes in detecting blending artifacts. From Figure 19, we can see that trained using self-blending techniques on real images to prevent overfitting, the SBI model's expert features show a strong correlation with fake labels, and its scoring effectively quantifies the extent of blending artifacts. Similarly, the incorporation of the *CFDA model* [41], an enhanced version of the SBI model, results in an additional performance boost, indicating that as the expert model's ability to capture blending features improved, the overall model's generalization capability also increases.

To explain **criteria-3**, we conducted additional experiments using non-uniform data distribution. Specifically, we created an extremely imbalanced dataset by removing a large portion of fake samples that do not contain the blending feature. As the imbalance increased, the model's performance degraded, and in extreme cases, it began to rely on shortcut solutions. In the **remove 95** and **remove 99.5** cases, we removed 95% and 99.5% of samples close to the real distribution, respectively, resulting in highly imbalanced datasets with mostly fake samples remaining.

Table 13: Performance Comparison of Different Models on Various Datasets. The **remove 95** and **remove 99.5** scenarios represent extreme cases of data imbalance by removing 95% and 99.5% of the samples near the real distribution, respectively.

| Varient | Celeb-DF-v2 | DFDCP | E4S | Facedancer | Fsgan | Inswap | Simswap | Average |
|---------|-------------|-------|-----|------------|-------|--------|---------|---------|
| remove 99.5 | 75.6 | 79.0 | 63.6 | 67.2 | 80.2 | 63.0 | 65.4 | 70.6 |
| remove 95 | 79.3 | 81.4 | 68.9 | 69.7 | 82.1 | 65.7 | 70.3 | 73.9 |
| CDFA+MLLM | **90.3** | **89.6** | **91.2** | **83.8** | **89.9** | **78.5** | **84.9** | **86.9** |

# H  Additional Analysis of Ablation Study

**Effects of Strong Feature Strengthening Module.**  We observed a substantial performance boost in the model after fine-tuning it with a dataset constructed using strong features, as evidenced by the leap from **Variant-1** to **Variant-5**. Remarkably, this significant improvement occurred even without the aid of specific feature detectors (WFS), underscoring the potency of strong feature utilization. This finding prompted us to investigate the underlying causes. We reexamined the feature capabilities of the pre-trained model (**Variant-1**) and compared them to those of the model enhanced with strong features (**Variant-5**). As illustrated in Figure 18, the majority of feature capabilities exhibited marked enhancement following the application of strong feature strengthening. Intriguingly, even some initially weaker features demonstrated noticeable improvement post-enhancement.

**Effect of Model Feature Assessment Module.**  Without Model Feature Assessment (MFA), identifying the model's strong features becomes impossible, and relying on weaker features undermines both the reliability of the model's explanations and its overall performance. To explore this, we allowed the model to construct the dataset using the entire list of forgery-related features rather than prioritizing strong ones. In **Variant-3**, which adopts this approach without enforcing strong feature use, performance lags significantly behind **Variant-5**, where MFA pinpoints and leverages strong features for fine-tuning, as evidenced in Table 14. To further investigate the absence of

Table 14: Ablation study regarding the effectiveness of each proposed module via cross-dataset evaluations. All models are trained on the FF++ c23 dataset and evaluated with metrics in the order of AUC ‖ AP ‖ EER (frame-level). The results show an incremental benefit in each module. We use ✓ to indicate the presence of a module and ✗ to indicate its absence.

| | Ours | | | CDF | | | DFD | | | DFDC | | | Simswap | | | Uniface | | | Avg. | | |
|---|---|---|---|---|---|---|---|---|---|---|---|---|---|---|---|---|---|---|---|---|---|
| # | MFA | SFS | WFS | AUC | AP | EER | AUC | AP | EER | AUC | AP | EER | AUC | AP | EER | AUC | AP | EER | AUC | AP | EER |
| 1 | ✗ | ✗ | ✗ | 52.1 | 68.2 | 48.7 | 69.8 | 95.2 | 36.4 | 57.8 | 59.9 | 44.6 | 64.0 | 64.1 | 40.4 | 65.5 | 65.6 | 39.0 | 61.8 | 70.6 | 41.8 |
| 2 | ✓ | ✗ | ✗ | 52.3 | 67.4 | 49.4 | 75.0 | 96.0 | 31.5 | 63.3 | 66.0 | 39.9 | 59.3 | 59.6 | 43.7 | 57.8 | 58.5 | 44.1 | 61.5 | 69.5 | 41.8 |
| 3 | ✗ | ✓ | ✗ | 79.0 | 88.3 | 28.9 | 88.9 | 98.7 | 18.0 | 77.8 | 81.9 | 28.9 | 82.0 | 84.0 | 25.9 | 82.3 | 84.8 | 25.2 | 82.0 | 87.3 | 25.6 |
| 4 | ✗ | ✗ | ✓ | 87.9 | 93.6 | 20.5 | 90.9 | 98.9 | 17.6 | 83.5 | 86.1 | 24.8 | 76.0 | 74.2 | 29.8 | 76.5 | 75.1 | 29.8 | 83.0 | 85.6 | 24.5 |
| 5 | ✓ | ✓ | ✗ | 83.2 | 90.5 | 24.6 | 91.4 | 99.0 | 15.8 | 79.2 | 82.1 | 27.6 | 83.3 | 85.0 | 24.8 | 84.5 | 86.2 | 22.4 | 84.9 | 88.5 | 23.0 |
| 6 | ✗ | ✓ | ✓ | 88.1 | 93.6 | 20.4 | 91.1 | 98.6 | 16.8 | 82.0 | 84.5 | 26.1 | 78.0 | 76.5 | 28.8 | 78.7 | 77.1 | 28.6 | 83.6 | 86.1 | 24.1 |
| 7 | ✓ | ✗ | ✓ | 87.3 | 93.2 | 21.0 | 90.2 | 98.7 | 18.0 | 82.0 | 82.1 | 26.6 | 76.7 | 75.0 | 29.5 | 77.2 | 75.8 | 29.0 | 82.6 | 84.9 | 24.8 |
| 8 | ✓ | ✓ | ✓ | **90.4** | **94.9** | **17.7** | **92.3** | **99.1** | **15.5** | **83.7** | **86.1** | **24.8** | **85.1** | **85.7** | **23.2** | **85.5** | **86.4** | **22.4** | **87.4** | **90.4** | **20.7** |

MFA, we introduced **Variant-6**, which integrates specific feature detectors (WFS) without MFA, and compared it to **Variant-8**, the full model with MFA, SFS, and WFS. The results clearly show Variant-6 underperforming Variant-8, highlighting a notable disadvantage when model assessment is omitted. These experiments Variant-3 versus Variant-5 and Variant-6 versus Variant-8 collectively demonstrate that assessing and utilizing strong features via MFA is critical for optimizing model effectiveness in deepfake detection. Strikingly, despite the pre-trained **Variant-1**'s limited detection ability, **Variant-2** reveals the model's feature capability by using a single MFA-identified strong feature (*e.g.*, distortion—present for fake, absent for real) for direct judgment without training. This underscores the model's strong feature-identification capability and the potential of tapping these features, which combinations in Variants 5 and 8 further amplify.

We evaluate the generalization performance of our model in cross-dataset evaluation scenarios through an ablation study involving several variants to systematically assess the contributions of different components. The variants include: **Variant-1**, a pre-trained MLLM LlaVa-1.5-7B[44] as baseline, without any feature strengthening or supplementation; **Variant-2**, which utilizes the single strongest feature identified by *Model Feature Assessment (MFA)* for deepfake detection; **Variant-3**, Assumes all features are strong and uses them to construct the dataset without prior evaluation or ranking, followed by fine-tuning the model; **Variant-4**, a simple test of the specific feature detectors ability, for this model can only output a single probability, as proposed in [41]; **Variant-5**, only use strong features to construct datasets after feature assessment, and use this dataset to fine-tune; **Variant-6**, Uses all strong features without feature assessment, integrating the *specific feature detectors (SFD)* to construct datasets, followed by fine-tuning the model.; and **Variant-7**, Constructs datasets solely using the *specific feature detectors (SFD)*, masking visual information to prevent strong feature leakage, followed by fine-tuning the model. **Variant-8**, Full model, integrating MFA, SFS, and WFS. The results, presented in Table 14, demonstrate incremental improvements across various datasets. For instance, Variant-8 achieves the highest average AUC (87.4%), AP (90.3%), and lowest EER (20.7%), highlighting the synergy of MFA, SFS, and WFS in optimizing deepfake detection performance.

**Feature Capability.** We also conduct a comparative study of feature capabilities before and after feature strengthening.

**Effect of inconsistent use of supplementary features in training and inference.** The model performs best when supplementary features are used consistently during both training and inference (average auc: *87.8*), indicating that these features significantly enhance performance. When supplementary features are omitted entirely from both stages, the performance drops (average auc: *0.83.3*), though it remains better than when features are used inconsistently. Specifically, when features are used during training but not inference, the performance suffers greatly (average auc: *76.6*), suggesting the model relies on these features and struggles without them at inference time. On the other hand, when features are introduced at inference but not used during training, the model achieves slightly better results (average auc: *82.5*), but it cannot fully leverage unseen features, showing the importance of using supplementary features consistently across both phases.

**Extension in new datasets.** Our model, trained on a mixture of datasets including FF++, showed improved overall performance when we added a new dataset without blending artifacts to the training process. This demonstrates that incorporating diverse datasets with supplementary features, even

Table 15: Impact of omitting supplementary features during Training and Adding Them During Inference, on Model Performance

| Variant | Celeb-DF-v2 | DFD | DFDC | DFDCP | DFR | WDF | FFIW | Avg. |
|---|---|---|---|---|---|---|---|---|
| no train + no infer | 83.2 | 91.4 | 79.2 | 82.0 | 93.7 | 76.8 | 76.6 | 83.3 |
| train + no infer | 76.5 | 84.7 | 71.0 | 72.0 | 89.5 | 71.3 | 71.3 | 76.6 |
| no train + infer | 81.7 | 90.6 | 79.1 | 81.3 | 92.6 | 74.7 | 77.3 | 82.5 |
| train + infer | **90.4** | **92.3** | **83.7** | **87.3** | **97.6** | **81.4** | **81.7** | **87.8** |

from different domains, as shown in Table 16 enhances the model's generalization and comprehensive performance.

Table 16: Comparison of Model Performance When Trained on FF++ Alone vs. FF++ and SRI Across Different Datasets

| Variant | CDF | DFDCP | DFDC | DFD | Uniface | e4s | Facedancer | Fsgan | Inswap | Simswap |
|---|---|---|---|---|---|---|---|---|---|---|
| Train FF++ | 90.4 | 87.3 | 83.7 | 92.3 | 85.5 | 91.2 | 83.8 | **91.1** | **81.2** | 85.1 |
| Train FF++ and SRI | **91.5** | **89.3** | **83.9** | **92.7** | **87.4** | **93.0** | **84.6** | 89.9 | 81.0 | **86.1** |

# I   Sample Showing

Here we present some failure cases of pre-trained MLLMs and then compare them with our results. See the **supplementary material** for more examples and details.

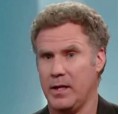

Label: real

Detection

**USER:** I want you to work as an image forensic expert for Deepfake faces. Check if the image has the artifact attribute listed in the following list and ONLY return the attribute number in this image. The artifact list is [1-asymmetric eye iris; 2-irregular glasses shape or reflection; 3-irregular teeth shape or texture; 4-irregular ears or earrings; 5-strange hair texture; 6-inconsistent skin texture; 7-inconsistent lighting and shading; 8-strange background; 9-weird hands; 10-unnatural edges].

**GPT: Attribute number**: Based on the provided list of artifacts, I reviewed the image and found that it displays the following attribute:**5-strange hair texture.** The texture of the hair appears somewhat artificial, a common characteristic of AI-generated images.

**USER:** This image is real, tell me the reason.

Explanation

**GPT:** The image you provided appears to be a real person, possibly from a television interview or public appearance. The reason it seems real is due to the natural lighting, human facial expressions, and other small details such as skin texture and natural positioning, which are characteristics of an authentic photograph of a person in a live setting. However, based on this image alone, I cannot confirm its exact source or context.

**USER:** Is this image is real or deepfake, tell me the reason.

Detection& Explanation

**GPT:** The image you provided appears to be a real person, possibly from a television interview or public appearance. The reason it seems real is due to the natural lighting, human facial expressions, and other small details such as skin texture and natural positioning, which are characteristics of an authentic photograph of a person in a live setting. However, based on this image alone, I cannot confirm its exact source or context.

Figure 20: Diagram showing how the GPT4o generates results for **detection**, **explanation**, and **detection with explanation**.

For LLaVa, we use the same prompts as GPT-4 to ensure fairness in the evaluation process. The robustness of Llava in these tasks is illustrated in Figure 21.

Table 17: Relationship between various facial features and deepfake detection (Part 1)

| No. | Question | Pretrain |
|---|---|---|
| 1 | Is the face color related to deepfake detection? | No |
| 2 | Are the eyes related to deepfake detection? | No |
| 3 | Are the facial features related to deepfake detection? | No |
| 4 | Is the nose contour related to deepfake detection? | No |
| 5 | Is face blurriness related to deepfake detection? | No |
| 6 | Is the skin tone related to deepfake detection? | No |
| 7 | Are the cheeks related to deepfake detection? | No |
| 8 | Is the skin tone pattern related to deepfake detection? | No |
| 9 | Is the placement of facial features related to deepfake detection? | No |
| 10 | Are the lips related to deepfake detection? | Yes |
| 11 | Is facial symmetry related to deepfake detection? | No |
| 12 | Is the lighting on the cheeks related to deepfake detection? | Yes |
| 13 | Is the facial lighting related to deepfake detection? | No |
| 14 | Are the shapes of facial features related to deepfake detection? | No |
| 15 | Is facial evenness related to deepfake detection? | Yes |
| 16 | Are the cheekbones related to deepfake detection? | No |
| 17 | Is the face layout related to deepfake detection? | No |
| 18 | Are the lip edges related to deepfake detection? | No |
| 19 | Is facial detail related to deepfake detection? | Yes |
| 20 | Is cheek smoothness related to deepfake detection? | No |
| 21 | Is the forehead shape related to deepfake detection? | No |
| 22 | Is face-background blending related to deepfake detection? | Yes |
| 23 | Is skin texture related to deepfake detection? | No |
| 24 | Are the eyelashes related to deepfake detection? | No |
| 25 | Are facial lines related to deepfake detection? | No |
| 26 | Is facial expression related to deepfake detection? | No |
| 27 | Is the nose shape related to deepfake detection? | No |
| 28 | Are color changes on the face related to deepfake detection? | No |
| 29 | Is the mouth shape related to deepfake detection? | No |
| 30 | Are the face edges related to deepfake detection? | No |
| 31 | Is facial rigidity related to deepfake detection? | No |
| 32 | Are sharp facial lines related to deepfake detection? | No |
| 33 | Is skin perfection related to deepfake detection? | No |
| 34 | Is forehead shininess related to deepfake detection? | Yes |
| 35 | Are sharp face edges related to deepfake detection? | Yes |
| 36 | Is skin smoothness related to deepfake detection? | No |
| 37 | Are eye details related to deepfake detection? | No |
| 38 | Are smooth facial lines related to deepfake detection? | No |
| 39 | Is lip texture related to deepfake detection? | Yes |
| 40 | Is forehead shine evenness related to deepfake detection? | No |
| 41 | Are the eyebrows related to deepfake detection? | No |
| 42 | Are unusual eye appearances related to deepfake detection? | No |
| 43 | Are facial transitions related to deepfake detection? | Yes |
| 44 | Is face color related to deepfake detection? | No |
| 45 | Is facial emotion exaggeration related to deepfake detection? | No |
| 46 | Is unusual face layout related to deepfake detection? | No |
| 47 | Are eye reflections related to deepfake detection? | No |
| 48 | Is skin texture roughness related to deepfake detection? | No |
| 49 | Is the jawline related to deepfake detection? | No |
| 50 | Is facial expression stiffness related to deepfake detection? | Yes |

Table 18: Relationship between various facial features and deepfake detection (Part 2)

| No. | Question | Pretrain |
|-----|----------|----------|
| 51 | Is nose texture related to deepfake detection? | No |
| 52 | Is skin shininess under the nose related to deepfake detection? | No |
| 53 | Is uneven facial sharpness related to deepfake detection? | Yes |
| 54 | Is facial blending related to deepfake detection? | No |
| 55 | Is facial lighting evenness related to deepfake detection? | No |
| 56 | Is nose bridge smoothness related to deepfake detection? | No |
| 57 | Is the hairline related to deepfake detection? | No |
| 58 | Is skin texture evenness related to deepfake detection? | No |
| 59 | Is facial feature balance related to deepfake detection? | No |
| 60 | Is facial symmetry related to deepfake detection? | No |
| 61 | Is forced facial expression related to deepfake detection? | No |
| 62 | Are the nostrils related to deepfake detection? | No |
| 63 | Are unnatural lip appearances related to deepfake detection? | No |
| 64 | Is partial skin smoothness related to deepfake detection? | No |
| 65 | Is lip texture related to deepfake detection? | No |
| 66 | Is lighting around the nose related to deepfake detection? | Yes |
| 67 | Are facial feature proportions related to deepfake detection? | Yes |
| 68 | Is skin smoothness around the nose related to deepfake detection? | No |
| 69 | Are soft facial creases related to deepfake detection? | No |
| 70 | Are teeth appearances related to deepfake detection? | No |
| 71 | Is neck-face transition related to deepfake detection? | No |
| 72 | Is skin tone variation related to deepfake detection? | No |
| 73 | Is face edge sharpness related to deepfake detection? | No |
| 74 | Is chin outline visibility related to deepfake detection? | No |
| 75 | Is facial lighting evenness related to deepfake detection? | Yes |
| 76 | Are ear details related to deepfake detection? | No |
| 77 | Is chin smoothness related to deepfake detection? | No |
| 78 | Are bright facial areas related to deepfake detection? | No |
| 79 | Is skin brightness near the mouth related to deepfake detection? | No |
| 80 | Are nostril appearances related to deepfake detection? | No |
| 81 | Are dimples related to deepfake detection? | Yes |
| 82 | Is jawline prominence related to deepfake detection? | No |
| 83 | Is under-eye texture related to deepfake detection? | No |
| 84 | Is facial blending related to deepfake detection? | Yes |
| 85 | Is chin shadow related to deepfake detection? | No |
| 86 | Are forehead shadows related to deepfake detection? | No |
| 87 | Is nose light reflection related to deepfake detection? | No |
| 88 | Is face-background transition related to deepfake detection? | No |
| 89 | Is forehead light reflection related to deepfake detection? | No |
| 90 | Are nose shadows related to deepfake detection? | No |
| 91 | Is lighting around the mouth related to deepfake detection? | No |
| 92 | Is neck smoothness related to deepfake detection? | No |
| 93 | Are face outlines related to deepfake detection? | No |
| 94 | Are face edges related to deepfake detection? | No |
| 95 | Are skin details related to deepfake detection? | No |
| 96 | Are under-eye shadows related to deepfake detection? | No |
| 97 | Are cheek shadows related to deepfake detection? | No |
| 98 | Are cheekbone appearances related to deepfake detection? | No |
| 99 | Is facial lighting related to deepfake detection? | No |
| 100 | Are facial wrinkle details related to deepfake detection? | No |

Table 19: Top Strong 50 Features

| Rank | Question | Pretrain | Strengthened |
|---|---|---|---|
| 1 | Is the face color unusual? | 0.6340 | 0.7486 |
| 2 | Is there something wrong with the eyes? | 0.6309 | 0.6320 |
| 3 | Do the facial features look oddly shaped? | 0.6292 | 0.6636 |
| 4 | Is the contour of the nose incorrect? | 0.6278 | 0.5817 |
| 5 | Is part of the face blurry? | 0.6231 | 0.7643 |
| 6 | Does the skin tone make the face look fake? | 0.6165 | 0.6479 |
| 7 | Is there something wrong with the cheek? | 0.6144 | 0.8082 |
| 8 | Are there strange patterns in the skin tone? | 0.6144 | 0.8075 |
| 9 | Are the face parts out of place? | 0.6130 | 0.7919 |
| 10 | Do the lips seem out of place or strangely shaped? | 0.6127 | 0.7408 |
| 11 | Is one side of the face uneven with the other? | 0.6123 | 0.7622 |
| 12 | Are there strange lighting spots on the cheeks? | 0.6111 | 0.8029 |
| 13 | Does the lighting change strangely on the face? | 0.6092 | 0.8006 |
| 14 | Are the shapes of the eyes, nose, or mouth unnatural? | 0.6054 | 0.6714 |
| 15 | Does the face look uneven or off? | 0.6048 | 0.6732 |
| 16 | Does the cheekbone appear too flat? | 0.6014 | 0.7406 |
| 17 | Does the face layout look wrong? | 0.5986 | 0.5422 |
| 18 | Are the edges of the lips too smooth? | 0.5979 | 0.6264 |
| 19 | Is part of the face lacking detail? | 0.5942 | 0.6843 |
| 20 | Are the cheeks too smooth? | 0.5934 | 0.6728 |
| 21 | Does the forehead look odd in shape? | 0.5911 | 0.7493 |
| 22 | Does the face mix poorly with the background? | 0.5902 | 0.6382 |
| 23 | Is the skin texture uneven? | 0.5861 | 0.7158 |
| 24 | Are the eyelashes missing or blurred? | 0.5857 | 0.6733 |
| 25 | Are the face lines uneven or changing in different areas? | 0.5826 | 0.6546 |
| 26 | Does the face lack expression? | 0.5822 | 0.6679 |
| 27 | Does the nose shape look odd? | 0.5812 | 0.5306 |
| 28 | Are the color changes on the face and skin sudden? | 0.5807 | 0.6643 |
| 29 | Does the mouth appear too flat? | 0.5775 | 0.6542 |
| 30 | Are the edges of the face too sharp? | 0.5774 | 0.8188 |
| 31 | Does the face appear too rigid? | 0.5770 | 0.7446 |
| 32 | Are the face lines too sharp? | 0.5761 | 0.7724 |
| 33 | Does the skin look too perfect, like it was edited? | 0.5755 | 0.5749 |
| 34 | Is the forehead too shiny? | 0.5737 | 0.8168 |
| 35 | Are the face edges too sharp? | 0.5720 | 0.8162 |
| 36 | Does the face skin look too smooth? | 0.5640 | 0.5306 |
| 37 | Are the eyes blurry or lacking detail? | 0.5636 | 0.5362 |
| 38 | Are the face lines too smooth? | 0.5549 | 0.5927 |
| 39 | Are the lips too smooth or lacking texture? | 0.5537 | 0.5475 |
| 40 | Is the forehead's shine uneven? | 0.5515 | 0.7208 |
| 41 | Are the eyebrows too dark or too light? | 0.5454 | 0.5075 |
| 42 | Do the eyes look odd? | 0.5433 | 0.5389 |
| 43 | Are transitions on the face poorly blended? | 0.5410 | 0.5854 |
| 44 | Do the face colors look strange? | 0.5382 | 0.6163 |
| 45 | Does the face show emotions that seem exaggerated? | 0.5355 | 0.6337 |
| 46 | Does the face layout look unusual? | 0.5344 | 0.5377 |
| 47 | Do the eyes have unnatural reflections? | 0.5323 | 0.6417 |
| 48 | Does the face have rough or uneven skin texture? | 0.5292 | 0.7973 |
| 49 | Does the jawline appear too sharp or unclear? | 0.5292 | 0.5017 |
| 50 | Does the facial expression look stiff? | 0.5289 | 0.5346 |

Table 20: Bottom 50 Weak Features

| Rank | Question | Pretrained | Strengthened |
|------|----------|-----------|--------------|
| 51 | Does the nose lack texture? | 0.5231 | 0.5111 |
| 52 | Is the skin too shiny under the nose? | 0.5223 | 0.7458 |
| 53 | Is the sharpness of the face uneven in parts? | 0.5214 | 0.5701 |
| 54 | Does the blending on the face look unnatural or uneven? | 0.5212 | 0.5151 |
| 55 | Is the lighting on the face strange or uneven? | 0.5200 | 0.5968 |
| 56 | Does the nose bridge appear too smooth? | 0.5172 | 0.5395 |
| 57 | Does the hairline seem unnatural? | 0.5148 | 0.5495 |
| 58 | Does the face skin texture look uneven? | 0.5144 | 0.5489 |
| 59 | Do the face parts look out of balance? | 0.5137 | 0.5887 |
| 60 | Are the facial features too symmetrical? | 0.5130 | 0.7300 |
| 61 | Does the facial expression look forced? | 0.5116 | 0.5562 |
| 62 | Are the nostrils hard to see? | 0.5115 | 0.6535 |
| 63 | Do the lips look unnatural? | 0.5110 | 0.5555 |
| 64 | Does the face skin look too smooth in some areas? | 0.5089 | 0.5287 |
| 65 | Do the lips lack natural texture? | 0.5083 | 0.5855 |
| 66 | Is the lighting around the nose inconsistent? | 0.5080 | 0.7257 |
| 67 | Do the sizes of the eyes, nose, and mouth seem off? | 0.5038 | 0.5275 |
| 68 | Does the skin around the nose look unnaturally smooth? | 0.5030 | 0.5309 |
| 69 | Are the facial creases too soft? | 0.5028 | 0.7943 |
| 70 | Do the teeth appear blurry or unrealistic? | 0.5028 | 0.5210 |
| 71 | Is the transition between the neck and the face not smooth? | 0.5026 | 0.5330 |
| 72 | Is the skin tone different in parts of the face? | 0.5023 | 0.5749 |
| 73 | Does the face lack sharpness around the edges? | 0.5021 | 0.5311 |
| 74 | Is the chin outline hard to see? | 0.5021 | 0.6160 |
| 75 | Is the lighting uneven on the face? | 0.5012 | 0.6351 |
| 76 | Are the details around the ears unclear? | 0.5010 | 0.6751 |
| 77 | Is the chin too smooth compared to the rest of the face? | 0.5010 | 0.5664 |
| 78 | Do the bright areas on the face seem odd? | 0.5007 | 0.5196 |
| 79 | Is the skin near the mouth unnaturally bright? | 0.5007 | 0.5930 |
| 80 | Are the nostrils blurry or unclear? | 0.5007 | 0.5125 |
| 81 | Are the dimples missing or poorly defined? | 0.5005 | 0.5000 |
| 82 | Is the jawline too pronounced or too faint? | 0.5000 | 0.5003 |
| 83 | Is the area under the eyes missing natural texture? | 0.5000 | 0.5111 |
| 84 | Is there blending on the face that looks edited? | 0.5000 | 0.5014 |
| 85 | Does the shadow under the chin seem unnatural? | 0.5000 | 0.5090 |
| 86 | Is the forehead missing natural shadows? | 0.5000 | 0.5000 |
| 87 | Does the light reflection on the nose look strange? | 0.5000 | 0.5049 |
| 88 | Are the transitions between the face and the background poorly blended? | 0.5000 | 0.5447 |
| 89 | Does the light reflection on the forehead look artificial? | 0.5000 | 0.5007 |
| 90 | Are there missing shadows around the nose? | 0.5000 | 0.5217 |
| 91 | Does the lighting around the mouth look unusual? | 0.5000 | 0.5301 |
| 92 | Does the neck look unnaturally smooth compared to the face? | 0.5000 | 0.6259 |
| 93 | Do the face outlines look off? | 0.5000 | 0.5247 |
| 94 | Do the edges around the face look unnatural? | 0.5000 | 0.5299 |
| 95 | Are the fine details on the skin missing? | 0.5000 | 0.5165 |
| 96 | Are the shadows under the eyes missing? | 0.5000 | 0.5000 |
| 97 | Are the cheeks lacking shadows? | 0.5000 | 0.5000 |
| 98 | Do the cheekbones appear unnaturally smooth? | 0.5000 | 0.5709 |
| 99 | Does the face appear overly lit in certain areas? | 0.5000 | 0.6758 |
| 100 | Are the wrinkles on the face lacking detail? | 0.5000 | 0.5014 |

Table 21: Questions list generated by Claude3.5-Sonnet (part1)

| No. | Question |
|---|---|
| 1 | Are there noticeable inconsistencies in facial symmetry? Return me yes or no |
| 2 | Does the skin texture appear artificially smooth or lacking natural details? Return me yes or no |
| 3 | Are the eyes misaligned or disproportionate? Return me yes or no |
| 4 | Is there unnatural blending between facial features and background? Return me yes or no |
| 5 | Do shadows and lighting appear inconsistent across the face? Return me yes or no |
| 6 | Are facial expressions unnatural or mechanically rigid? Return me yes or no |
| 7 | Does the hairline show signs of artificial blending? Return me yes or no |
| 8 | Are there visible artifacts or glitches in the image? Return me yes or no |
| 9 | Do reflections in the eyes match the environment? Return me yes or no |
| 10 | Is there proper alignment of facial features? Return me yes or no |
| 11 | Does the skin show natural imperfections and pores? Return me yes or no |
| 12 | Are teeth shapes and alignment realistic? Return me yes or no |
| 13 | Is there consistent image quality across all facial areas? Return me yes or no |
| 14 | Do facial proportions follow natural human anatomy? Return me yes or no |
| 15 | Are shadows cast appropriately based on lighting? Return me yes or no |
| 16 | Does facial hair follow natural growth patterns? Return me yes or no |
| 17 | Is there proper depth and dimension to facial features? Return me yes or no |
| 18 | Are color tones consistent throughout the face? Return me yes or no |
| 19 | Do glasses and accessories appear properly attached? Return me yes or no |
| 20 | Is there natural variation in skin texture? Return me yes or no |
| 21 | Are facial contours anatomically correct? Return me yes or no |
| 22 | Does the head size match body proportions? Return me yes or no |
| 23 | Is there appropriate detail in fine features? Return me yes or no |
| 24 | Are transitions between features naturally blended? Return me yes or no |
| 25 | Do facial movements appear fluid and natural? Return me yes or no |
| 26 | Are ear shapes and positions symmetrical? Return me yes or no |
| 27 | Do eyebrows have natural hair patterns? Return me yes or no |
| 28 | Is there consistent resolution between face and background? Return me yes or no |
| 29 | Are nose contours anatomically accurate? Return me yes or no |
| 30 | Does makeup application appear natural? Return me yes or no |
| 31 | Are facial wrinkles and lines age-appropriate? Return me yes or no |
| 32 | Do eyelashes appear realistic and properly attached? Return me yes or no |
| 33 | Is there natural skin coloration variation? Return me yes or no |
| 34 | Are facial highlights consistent with lighting? Return me yes or no |
| 35 | Do lips have natural texture and color? Return me yes or no |
| 36 | Is there proper depth in eye sockets? Return me yes or no |
| 37 | Are facial moles and marks naturally placed? Return me yes or no |
| 38 | Do teeth have individual characteristics? Return me yes or no |
| 39 | Is there natural asymmetry in facial features? Return me yes or no |
| 40 | Are skin pores visible where expected? Return me yes or no |
| 41 | Do facial muscles move naturally? Return me yes or no |
| 42 | Is there consistent focus across the image? Return me yes or no |
| 43 | Are shadows under facial features natural? Return me yes or no |
| 44 | Do earrings and jewelry sit naturally? Return me yes or no |
| 45 | Is there proper skin subsurface scattering? Return me yes or no |
| 46 | Are facial proportions consistent in different angles? Return me yes or no |
| 47 | Do eye corners have natural creases? Return me yes or no |
| 48 | Is there natural variation in lip texture? Return me yes or no |
| 49 | Are facial hair shadows realistic? Return me yes or no |
| 50 | Do glasses cast appropriate shadows? Return me yes or no |

Table 22: Questions list generated by Claude3.5-Sonnet (part2)

| No. | Question |
|-----|----------|
| 51 | Is there natural skin translucency? Return me yes or no |
| 52 | Are facial expressions emotionally consistent? Return me yes or no |
| 53 | Do neck muscles align naturally? Return me yes or no |
| 54 | Is there proper depth in smile lines? Return me yes or no |
| 55 | Are eye reflections consistent with scene lighting? Return me yes or no |
| 56 | Do facial features maintain proportion when moving? Return me yes or no |
| 57 | Is there natural skin aging present? Return me yes or no |
| 58 | Are hair strands individually visible? Return me yes or no |
| 59 | Do facial veins appear natural where visible? Return me yes or no |
| 60 | Is there consistent skin tone across transitions? Return me yes or no |
| 61 | Are nostril shapes symmetrical? Return me yes or no |
| 62 | Do ears have natural internal structure? Return me yes or no |
| 63 | Is there proper depth in nasolabial folds? Return me yes or no |
| 64 | Are eye bags and circles age-appropriate? Return me yes or no |
| 65 | Do facial piercings sit naturally? Return me yes or no |
| 66 | Is there natural variation in beard density? Return me yes or no |
| 67 | Are lip lines naturally defined? Return me yes or no |
| 68 | Do cheekbones have natural contours? Return me yes or no |
| 69 | Is there proper temple definition? Return me yes or no |
| 70 | Are eye whites naturally textured? Return me yes or no |
| 71 | Do facial scars appear authentic? Return me yes or no |
| 72 | Is there natural jaw definition? Return me yes or no |
| 73 | Are facial dimples naturally placed? Return me yes or no |
| 74 | Do eyebrow hairs have direction variation? Return me yes or no |
| 75 | Is there proper chin definition? Return me yes or no |
| 76 | Are facial freckles naturally distributed? Return me yes or no |
| 77 | Do eyelids have natural creases? Return me yes or no |
| 78 | Is there consistent skin shininess? Return me yes or no |
| 79 | Are facial tattoos properly embedded? Return me yes or no |
| 80 | Do smile lines appear natural? Return me yes or no |
| 81 | Is there proper forehead texture? Return me yes or no |
| 82 | Are eye corners naturally aged? Return me yes or no |
| 83 | Do facial muscles show proper definition? Return me yes or no |
| 84 | Is there natural lip symmetry? Return me yes or no |
| 85 | Are ear lobes naturally shaped? Return me yes or no |
| 86 | Do facial shadows have color variation? Return me yes or no |
| 87 | Is there proper nose bridge definition? Return me yes or no |
| 88 | Are facial pores consistently sized? Return me yes or no |
| 89 | Do eyebrows have natural thickness variation? Return me yes or no |
| 90 | Is there natural skin elasticity? Return me yes or no |
| 91 | Are facial creases movement-appropriate? Return me yes or no |
| 92 | Do teeth have natural translucency? Return me yes or no |
| 93 | Is there proper cheek coloring? Return me yes or no |
| 94 | Are eye bags naturally shadowed? Return me yes or no |
| 95 | Do facial features maintain proper scale? Return me yes or no |
| 96 | Is there natural skin undertone? Return me yes or no |
| 97 | Are facial expressions muscle-consistent? Return me yes or no |
| 98 | Do wrinkles have proper depth? Return me yes or no |
| 99 | Is there natural facial bone structure? Return me yes or no |
| 100 | Are skin textures consistently detailed? Return me yes or no |

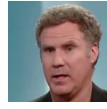

Label: real

**Detection**

USER: I want you to work as an image forensic expert for Deepfake faces. Check if the image has the artifact attribute listed in the following list and ONLY return the attribute number in this image. The artifact list is [1-asymmetric eye iris; 2-irregular glasses shape or reflection; 3-irregular teeth shape or texture; 4-irregular ears or earrings; 5-strange hair texture; 6-inconsistent skin texture; 7-inconsistent lighting and shading; 8-strange background; 9-weird hands; 10-unnatural edges].

**LLaVa: 1**

USER: This image is real, tell me the reason.

**Explanation**

LLaVa: The reason for the image being real is related to the subject matter depicted in the image, which is a man with a beard and mustache. The man appears to be the famous actor, John Wayne, known for his distinctive facial appearance in movies. The image is a close-up, and John Wayne's facial features are clearly visible and recognizable. It's not a photo-shopped or digitally altered image. The context of the image is John Wayne talking to someone, which is also consistent with his well-known roles.

USER: Is this image is real or deepfake, tell me the reason.

**Detection & Explanation**

LLaVa: The image is real, as it is an authentic photo of John Wayne, a famous actor known for his distinctive beard and mustache. The close-up view of John Wayne's face in the image allows for clear recognition of his facial features, making it a genuine representation of him.

Figure 21: Diagram showing how the LLaVa generates results for **detection**, **explanation**, and **detection with explanation**.

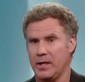

Label: real

**Detection**

USER: Is this image real or fake?

OURS: This image is real. The image appears to be real, as the man's facial features, such as his nose, eyes, and mouth, are well-defined and not distorted. The skin tone appears natural, and there are no signs of blurriness or unusual lighting. The man's facial expressions and overall appearance seem genuine, making it likely that the image is a true representation of the man. And this image does not contain any blending artifacts.

USER: This image is real, tell me the reason.

**Explanation**

OURS: The image of the man's face appears to be real, as it displays a well-defined facial structure with clear skin, prominent eyes, and a nose that is neither overly smooth nor has flat cheekbones. The man's facial features are well-defined, and there is no noticeable blurriness or distortion in the image. The overall appearance of the man's face suggests that it is a genuine representation of the individual. And this image does not contain any blending artifacts.

USER: Is this image real or fake?

**Detection & Explanation**

OURS: This image is real. The image appears to be real, as the man's facial features, such as his nose, eyes, and mouth, are well-defined and not distorted. The skin tone appears natural, and there are no signs of blurriness or unusual lighting. The man's facial expressions and overall appearance seem genuine, making it likely that the image is a true representation of the man. And this image does not contain any blending artifacts.

Figure 22: Diagram showing how the ours model generates results for **detection**, **explanation**, and **detection with explanation**.

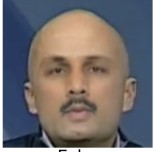

Fake

**USER:** Is this image real or fake? Tell the reason in detail?

**MLLM:** The image is real, featuring a bald man wearing a blue shirt and a white collar. He is talking and has a mustache. The image is not photoshopped. The man appears to be a news reporter or a presenter, and his expression suggests that he is engaged in a discussion or interview. The image is a genuine representation of the man and his attire.

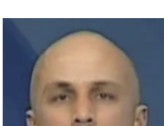

Fake

**USER:** This is fake image, tell the reason in detail.

**MLLM:** The image is a fake or manipulated picture of a bald man with a goatee, likely created for entertainment or artistic purposes. The reason for this manipulation could be to create a humorous or unconventional visual effect, to make a statement, or to generate curiosity and discussion. The image may have been created using photoshop software, resulting in the unusual combination of a bald man with a goatee. This manipulation goes beyond the typical representation of a bald man, as it adds a distinctive element to the image that is not commonly seen in real life.

Figure 23: Bad sample of Pre-trained model (part1)

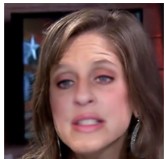

**USER:** Is this image real or fake? Tell the reason in detail.

**MLLM:** The image is real, as it shows a woman with blue eyes and red lips. The image also features a TV in the background, which is an indication that the setting is likely a TV studio or a live broadcast. The presence of the TV and the woman's makeup suggest that this is a real, live situation, rather than a fake or manipulated image.

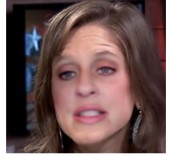

Fake

**USER:** Why this image is fake?

**MLLM:** The image is fake because the person's face appears to be photoshopped, This can be inferred from the fact that her eyes do not appear to be looking at the camera, and the lighting and background are not consistent. Additionally, the woman's face is distorted, and she has fake eyelashes and a blurry face. The face manipulation creates an unrealistic and artificial appearance, making it clear that the image is not a genuine photograph.

Figure 24: Bad sample of Pre-trained model (part2)

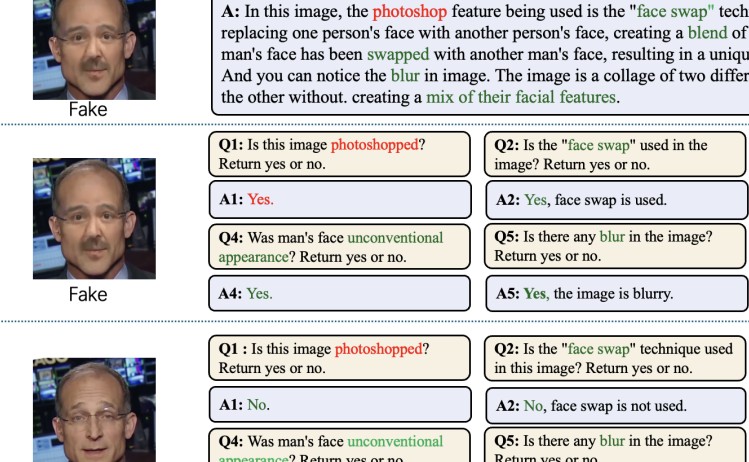

Figure 25: Bad sample of Pre-trained model (part3)

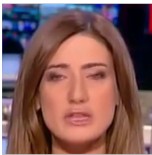

**USER:** Is this image real or fake? Tell the reason in detail?

**Pretrained MLLM:** This image appears real. It shows a woman speaking into a camera with her mouth wide open, possibly in an interview setting. Deepfake images often involve more discrepancies in facial features, lighting, and overall content when compared to a regular image with a genuine, real-world scenario. The image seems to capture a natural moment, so it does not suggest any deepfake content as a possibility.

**USER:** This image is fake, tell me the reasons.

**Pretrained MLLM:** The image is fake because it is a manipulated photo of a woman with a different background. The woman's face is placed onto a different body, creating an unnatural and unrealistic appearance. This manipulation is commonly done for entertainment, artistic purposes, or to create a humorous effect. The image does not depict a real-life scenario, and the altered elements make it a fake representation of a person.

**USER:** Is this image real or fake?

**$\chi^2$-DFD :** This image is fake. The image appears to be a deepfake as the woman's face appears to be distorted, with an unusual layout and unnatural skin tone. The facial features seem to be mismatched, and the overall appearance of the face is not consistent with a real person. And image show the obvious blending artifact.

Figure 26: Comparison between Pre-trained LLaVa and ours

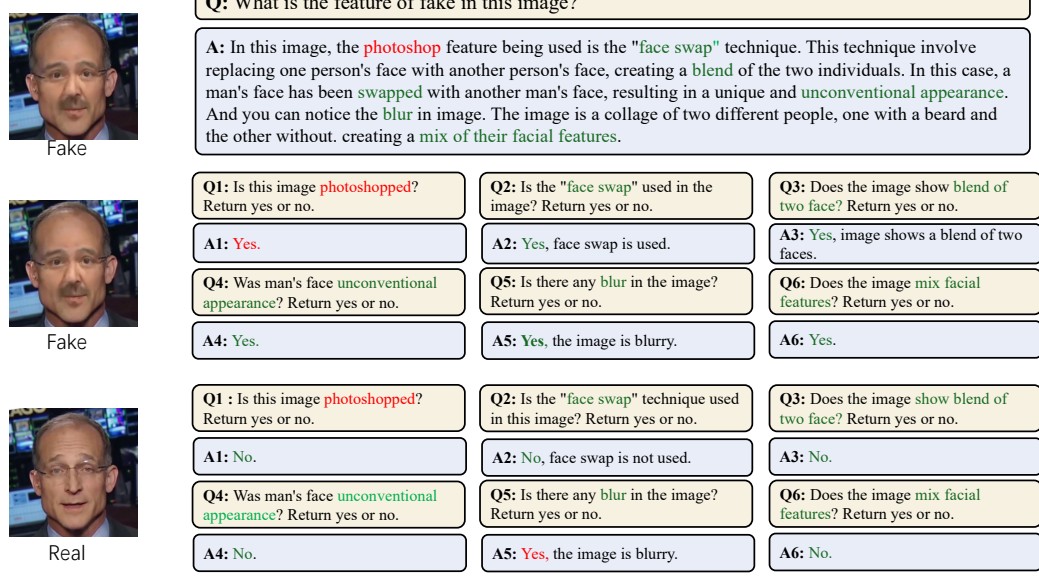

Figure 27: Feature related questions

