# OpenReview forum: "X2-DFD: A framework for explainable and extendable Deepfake Detection"
_NeurIPS.cc/2025/Conference — NeurIPS 2025 poster_

### Official Review · Reviewer_1fa1 · 2025-06-29

**Clarity:** 3
**Significance:** 2
**Originality:** 2
**Rating:** 4
**Confidence:** 4

**Summary:**

This paper tackles the problem of deepfake detection with the help of an MLLM by funetuning it with a self-constructed new dataset. The dataset constructing process consists of 2 stages. To construct the dataset, for a given sample, it first evaluates and ranks forgery-related features (as strong and weak features)  based on their detectability. Based on the ranking results, they further suggest directly using the strong feature as classification prompts, while for the weak features, it includes an additional prompts related to its blending factor. After finetuning the MLLM on the newly contructed dataset via LoRA, this method shows leading performance in different datasets.

**Questions:**

See weakness

**Ethical Concerns:**

["NO or VERY MINOR ethics concerns only"]

**Final Justification:**

Most of my concerns are addressed. After reading comments from other reviewers, I change my rating to positive.

**Limitations:**

The dataset construction process in the proposed framework relies on human verification to ensure the reliability of identified weak features, which can be labor-intensive and time-consuming for large datasets. This dependency on human intervention may introduce scalability challenges and could potentially limit the framework’s performance due to human subjectivity or errors in feature validation.

**Quality:**

3

**Strengths And Weaknesses:**

Pros:

1. This method focuses on generating reliable explanations by prioritizing features the model can reliably detect, which is relatively new in the field;

2. The manuscript is overall well-written, the pipeline figures clear and easy to understand.

Cons:

1. The context don't elaborate on how to group features as strong or weak, is there a selected threshold for the computed balance accuracy? If yes, there should be ablation study regarding this matter.

2. The current weak feature supplementing module relies on the blending factor as additional classification clue, why specifically use this? There are also other low-level clues in previous works that can be used for classification (lightening or shadow as shown in Fig. 2). Would it be better to use others or combine them? More justifications are required in here.

3. How about the results of using MLLMs finetuning only with the original dataset for comparison?

4. It is suggested to put the zero-shot acc. of the MLLM in the manuscript instead of the supplementary material.

---

> ### Author Rebuttal · Authors · 2025-07-31
>
> Dear Reviewer 1fa1，
>
> We are sincerely grateful for the reviewer's feedback. We are encouraged by your recognition of the **novelty of our approach to generating reliable explanations** and for their positive feedback on the clarity of the manuscript and the figures.
>
> ---
>
> >**Q1. Methodological Question 1 (Threshold for Features):** The context doesn't elaborate on how to group features as strong or weak, is there a selected threshold for the computed balance accuracy? If yes, there should be ablation study regarding this matter.
>
>
> **R1:** Thanks for your insightful question! In this work, we do not use a fixed accuracy threshold to group features. Instead, our approach is based on **relative ranking**. Specifically,
>
> - We rank all potential features by their balance accuracy performance and designate the **top-K as 'strong'** and the remainder as 'weak'. We believe this is a more robust method, as the concept of a 'strong' feature is relative to the model's capabilities.
>
> - Following your suggestion, we conducted an ablation study on the number of strong features, $K$, to analyze its impact on both generalization (AUC) and explanation quality (evaluated by GPT-4o, see Appendix D.2).
>
> - As shown in **Table 1**, our analysis reveals a clear trade-off:
>     * A **small $K$** (e.g., $K=25$) can be suboptimal because the feature set is not comprehensive enough.
>     * A **large $K$** (e.g., $K=100$), which includes many weak features, is also detrimental. These less distinguishable features act as **noise**, distracting the model's attention and causing it to generate unreliable explanations. This ultimately degrades detection performance.
>
> The optimal performance is achieved at **$K=50$**, which strikes the most effective balance between feature comprehensiveness and reliability.
>
> **Table 1: MLLM ablation study on the number of strong features ($K$). The evaluation metric is AUC.**
> | **Model Configuration** | **CDF** | **Uniface** | **HPS-Diff** | **Avg** | **GPT4o Eval** |
> | :--- | :---: | :---: | :---: | :---: | :---: |
> | X2DFD (Top-K=25) | 83.0 | 84.3 | 87.1 | 84.8 | 2.91 |
> | **X2DFD (Top-K=50)** | **83.2** | **84.5** | **88.7** | **85.5** | **3.02** |
> | X2DFD (Top-K=75) | 81.9 | 83.2 | 84.6 | 83.2 | 2.77 |
> | X2DFD (Top-K=100) | 79.0 | 82.3 | 83.6 | 81.6 | 2.63 |
>
> ---
>
> >**Q2.** How about the results of using MLLMs finetuning only with the original dataset for comparison?
>
> **R2:** Thank you for your insightful question, follw your suggestion we add the experiment for comparison:
>
> * **MLLM Alone (Tuning on Prioritizing Features):** Fine-tuning the MLLM using our curated set of high-quality, discriminative features achieves an average AUC of **85.5%**. This approach provides the model with meaningful data that improves its detection capabilities.
>
> * **No explanation (Tuning on Origin):** Fine-tuning the MLLM on only the original dataset's binary **real/fake** labels results in a lower average AUC of **82.3%**.
>
> This comparison clearly demonstrates that our strategy of strategically selecting and using high-quality features for fine-tuning leads to a substantial improvement in performance over the standard baseline approach.
>
>
> **Table 2: Comparison of Fine-tuning Methods. The evaluation metric is AUC.**
> | **Model Configuration** | **CDF** | **Uniface** | **HPS-Diff** | **Avg** |
> | :--- | :---: | :---: | :---: | :---: |
> | **MLLM Alone** | **83.2** | **84.5** | **88.7** | **85.5** |
> | **No explanation**| 80.3 | 81.9 | 84.7 | 82.3 |
>
>
>
> ---
>
>
> >**Q3.** The current weak feature supplementing module relies on the blending factor as additional classification clue, why specifically use this? There are also other low-level clues in previous works that can be used for classification (lightening or shadow as shown in Fig. 2). Would it be better to use others or combine them? More justifications are required in here.
>
>
>
> **R3:** Thank you for this insightful question. Our use of **blending clues** in the manuscript was intended as a **primary and clear-cut example** to illustrate our core thesis: MLLMs are strong generalists that benefit from specialist detectors (SFDs) for fine-grained cues they often miss. Actually, the SFD used in our framework can be implemented by any specific expert.
>
> To empirically validate the flexibility and power of our framework beyond just blending artifacts, and following the excellent suggestions from you and other reviewers, we have integrated an additional SFD that targets a completely different, yet highly relevant, low-level artifact: **diffusion traces**.
>
> This new experiment demonstrates two key points:
> - Our framework's performance is not tied to any single artifact; it is **truly extendable**.
> - Combining multiple SFDs targeting different artifacts results in a **synergistic effect**, leading to superior and more comprehensive detection capabilities.
>
>
> As the results in **Table 4** clearly show, integrating a diffusion-specialized SFD alongside the original blending-based SFD leads to a massive leap in overall performance, boosting the average AUC from 84.8% to **89.9%**.
>
>
>
> In summary, the **choice of blending was illustrative, not restrictive**. The core contribution is our **pluggable architecture**, which is artifact-agnostic and designed to leverage any number of specialist detectors. We thank you for prompting this clarification and will include these compelling new results in our revised manuscript.
>
>
>
> **Table 3: Performance with multiple SFDs, evaluation metric is AUC.**
>
> | Model | CDF (Blending) | SimSwap (No Blend) | Diff-FE (Diffusion) | Diff-I2I (Diffusion) | **Average** |
> | :--- | :---: | :---: | :---: | :---: | :---: |
> | **X2DFD (w/ Blending SFD)** | **90.4** | 85.1 | 82.1 | 81.7 | 84.8 |
> | **X2DFD (w/ Ble & Diff SFDs)** | 90.3 | **88.5** | **92.1** | **88.6** | **89.9** |
>
>
> ---
>
>
>
>
> >**Q4.** The dataset construction process in the proposed framework relies on human verification to ensure the reliability of identified weak features, which can be labor-intensive and time-consuming for large datasets. This dependency on human intervention may introduce scalability challenges and could potentially limit the framework’s performance due to human subjectivity or errors in feature validation.
>
> **R4:** Thank you for raising this important question! Indeed, there is a trade-off between reliability and efficiency in human verification during the finetuned dataset construction process. While human annotation helps reduce hallucinations in LLM outputs and improves dataset quality, it can be costly and time-consuming—especially at scale. This challenge has widely been recognized across the LLM community.
>
> To address this, we plan to train a preference-based MLLM to emulate human judgment, enabling automatic verification without ongoing human intervention. This approach aligns with common practice in scalable dataset curation and offers a practical path toward efficient, high-quality data generation.
>
>
>
> ---
>
>
>
> >**Q5. Presentation Issue:** The zero-shot accuracy of the MLLM should be moved from the supplementary material to the main manuscript for better visibility.
>
> **R5:** We appreciate your suggestion and will incorporate it into the revised version of the manuscript.

---

> > ### Comment · Reviewer_1fa1 · 2025-08-07
> > **Thanks for the rebuttal**
> >
> > The authors provide a good rebuttal. I thus change my rating to positive. It'll be good if these new results can be included in their future version.

---

> > > ### Author Response · Authors · 2025-08-07
> > > **Thanks for your reply**
> > >
> > > Thank you for your thoughtful feedback, and we are happy to see that our response has addressed your concerns!
> > >
> > > We greatly appreciate your suggestions and will be sure to incorporate these updates into the revised manuscript.
> > >
> > > Thank you again for your time and constructive review.

---

### Official Review · Reviewer_6sJc · 2025-07-01

**Clarity:** 4
**Significance:** 3
**Originality:** 3
**Rating:** 5
**Confidence:** 5

**Summary:**

This paper introduces X2-DFD, a novel framework for explainable and extendable deepfake detection that leverages Multimodal Large-Language Models (MLLMs). The core idea is to strategically utilize MLLM strengths while compensating for their inherent weaknesses, significantly enhancing both the detectability and explainability of deepfake images. The framework's modular design allows for seamless integration with future, more advanced MLLMs and Specific Feature Detectors (SFDs), ensuring its continuous improvement and scalability in the face of rapidly evolving deepfake challenges.

**Questions:**

（1）How will the training dataset be expanded to include newer deepfake types (e.g., from diffusion models) to test the framework's real-world applicability? (Refers to Weakness 1)

（2）What are the plans for simultaneously integrating multiple SFDs, and what are the expected benefits or challenges? (Refers to Weakness 2)

If both questions are addressed well, I will consider increasing the score.

**Ethical Concerns:**

["NO or VERY MINOR ethics concerns only"]

**Final Justification:**

After reading the rebuttal, I believe the authors have clarified some of my concerns. I think the paper offers some valuable insights for the community.

**Limitations:**

X2-DFD mainly focuses on face-swapping deepfakes; its effectiveness with full image generation (e.g., diffusion models) is less explored. Like the author mentioned implementation focuses solely on static image detection, the framework could expand to other data types like audio or audio-visual forgery. However, for face forgery images discussed in paper, the framework is very comprehensive.

**Quality:**

4

**Strengths And Weaknesses:**

Strengths

(1)Seamless SFD Integration: X2-DFD's modular design allows for the seamless integration of Specific Feature Detectors (SFDs). This modularity is crucial in the rapidly developing field of generative AI, ensuring the framework's continuous improvement and adaptability to new deepfake techniques.

(2)Comprehensive Experiments and Clear Explanations: The paper presents comprehensive experiments covering both deepfake detection and explanation.

(3)Novel and potential Core Idea: The methodology behind X2-DFD is highly reasonable and innovative. The approach of leveraging MLLM strengths while compensating for their weaknesses offers significant potential for future framework expansion and maybe a fundamental solution for both deepfake detection and explanation.

Weaknesses

(1)Limited Scope for Forgeries: The current framework primarily demonstrates strong performance in face-swapping. However, the exploration of more advanced techniques like Entire Face Synthesis (EFS) forgeries, particularly those generated by diffusion models, is relatively limited. Expanding the training dataset to include these latest forgery types would be beneficial.

(2)Underexplored Simultaneous Use of Multiple SFDs: While the authors comprehensively discuss SFD integration, the paper does not explore or discuss scenarios involving the simultaneous use of multiple SFDs. Integrating multiple "expert" SFDs concurrently could offer a more robust solution for detecting and explaining diverse types of fakes.

(3)Minor Writing Issue: In Section 5, "Ablation Study and Analysis," the phrasing of the question "How to ensure WFS module" should be rephrased for clarity.

---

> ### Author Rebuttal · Authors · 2025-07-31
>
> Dear Reviewer 6sJc,
>
> We sincerely appreciate your constructive feedback and are encouraged by your recognition of our work's strengths, including its **seamless SFD integration, comprehensive experiments, and novel core idea**. We will now address your comments in detail.
>
> ---
>
>
> >**Q1** How will the training dataset be expanded to include newer deepfake types (e.g., from diffusion models) to test the framework's real-world applicability?
>
>
> **R1:** Thank you for this critical question regarding the framework's generalization to new deepfake types. To directly investigate this, we conduct an experiment focused on expanding the training data.
>
> - We first establish a baseline model trained **only on the FF++ dataset**. We then expand this training set by **adding 1,000 images from two different modern, diffusion-based datasets** to assess the impact on generalization:
>     - **FF++ & XDXL:** adding entire face synthesis (EFS) images generated by the SDXL diffusion model.
>     - **FF++ & Diffface:** adding images generated by a diffusion-based face-swapping method.
>
>
> - The results in **Table 1** demonstrate that simply adding a small number of new examples to the training data yields only limited improvement in generalization. We expand our latest tests cover the four **most common types of facial forgeries: Face-Swap (FS), Image-to-Image (I2I), Text-to-Image (T2I), and Face-Editing (FE)**. The average performance only increased from 87.3% to 87.7%.
>
>
> **Table 1: Evaluating generalization by extending training datasets, evaluation metric is AUC.**
> | Training Set| CDF | Simswap | Diff-FE | Diff-FS | Diff-I2I | Diff-T2I |**Average** |
> | :--- | :---: | :---: | :---: | :---: | :---: | :---: |:---:|
> | FF++ (Baseline) | **90.4** | 85.1 |82.1 |**96.0** |81.7 | 88.3| 87.3 |
> | FF++ & XDXL |88.4 |83.8 | **85.3** | 92.4 | **84.7** | 87.5 | 87.0 |
> | FF++ & Diffface | 89.3 | **85.3** |84.3 |95.4 |83.3 | **88.6**| **87.7** |
>
>
>
> ---
>
>
> >**Q2.** Underexplored Simultaneous Use of Multiple SFDs: While the authors comprehensively discuss SFD integration, the paper does not explore or discuss scenarios involving the simultaneous use of multiple SFDs. Integrating multiple "expert" SFDs concurrently could offer a more robust solution for detecting and explaining diverse types of fakes.
>
> **R2:** Thank you for this excellent and insightful question. You have identified the core principle of extendability in our framework, and we conduct experiments to validate this exact direction **integrating multiple SFDs**.
>
> - To test this, we build directly on our previous experiments, using the FF++ & Diffface dataset as our training set. We then compare two models:
>     - Single SFD: X2DFD using only the blending-based SFD.
>
>     - Multiple SFDs: X2DFD integrating two SFDs simultaneously—the **original blending-based SFD and a new, diffusion-based feature detector.**
>
>     - The results, shown in **Table 2** below, clearly highlight the superiority of the multi-SFD approach. By integrating multiple specialized SFDs, the model's average performance boosts **from 87.7% to 91.5%**. This experiment demonstrates two advantages of framework extensibility：
>         - Our framework can **simultaneously expand multiple SFDs.**
>         - **By extending the corresponding SFD model, supplementing the framework's weakness**, and the overall performance of the framework has been improved.
>
> **Table 2: Performance on integrating multiple SFDs, evaluation metric is AUC.**
> | Model | CDF | Simswap | Diff-FE | Diff-FS | Diff-I2I | Diff-T2I |**Average**|
> | :--- | :---: | :---: | :---: | :---: | :---: | :---: |:---:|
> | **Xception** |73.7 |77.6 | 58.6 | 85.9 | 56.8 | 62.4 |69.2
> | single SFD  | 89.3 | 85.3 |84.3 |95.4 |83.3 | 88.6| 87.7 |
> | **ble&Diff (w/ New SFD)** | **90.3** | **88.5** | **92.1** | **97.2**| **88.6** | **92.2** | **91.5** |
>
> ---
>
> >**Q3.** What are the plans for simultaneously integrating multiple SFDs, and what are the expected benefits or challenges?
>
> **R3:** Thank you for this excellent question, which touches upon the core vision for our framework's extendability and future evolution. Our response covers our current achievements, the long-term benefits, the primary challenges, and our plans to address them.
>
> - **Current achievement: successful integration of multiple SFDs.**
>     - We are pleased to report that we've already **successfully implemented and validated the simultaneous integration of multiple Specialized Forgery Detectors (SFDs)**.
>     - As shown in our experimental results in **Table 2**, combining detectors designed for different artifact types (e.g., blending-based and diffusion-based forgeries) **leads to a significant performance improvement**.
>     - This confirms the immediate benefit of our multi-SFD approach, proving that a synergistic combination of specialized "experts" can indeed elevate overall detection capabilities. This initial success forms the robust foundation for our future advancements.
>
>
> - **Future plan: automated and high-Performance integration**
>     - Our plan is to evolve the current integration process into a **more automated and higher-performance** intelligent system. We aim to develop a sophisticated automated framework that can not only rapidly evaluate and select new SFDs as they emerge but also **dynamically optimize their combinations**.
>     - This means moving beyond a static aggregation of detectors to a system that intelligently understands how different SFDs interact and can configure them for optimal performance against evolving forgery techniques. This intelligent automation will be crucial for maintaining state-of-the-art detection capabilities in a rapidly changing threat landscape.
>
> - **Core benefit: tackling diverse and complex forgeries in real-world Scenarios**
>
>     - A key benefit of integrating multiple SFDs is our enhanced ability to **address the increasingly diverse and complex types of forgeries encountered in real-world scenarios**. Forgery techniques are not monolithic; they vary widely from subtle pixel manipulations to sophisticated AI-generated deepfakes, often combining multiple methods. By leveraging SFDs that specialize in detecting different artifact signatures.
>
> - **Core challenge: feature conflict and coupling in a multi-expert system**
>
>     - The core challenge we anticipate as the number of integrated SFDs increases relates to **navigating potential feature conflict and interdependencies among these "experts."**
>     - **Adapting SFDs for greater specialization.** Another significant challenge might involve modifying individual SFDs to become even more vertical and specialized.
>         - As we integrate more detectors, we may find that some SFDs are not sufficiently granular in their focus. Reworking these existing SFDs to detect even finer, more specific types of forgery artifacts—perhaps by refining their training data or architectural biases—could be a challenge.
>         - It requires a deep understanding of each forgery type's unique fingerprint and the ability to surgically enhance SFDs without compromising their overall performance or introducing new biases.
>
>
> ---
>
> >**Q4.** Minor Writing Issue.
>
> **R4:** Thank you for your valuable comments. We will update the corresponding position in the revised manuscript.

---

> > ### Comment · Reviewer_6sJc · 2025-08-07
> >
> > Thanks to the author for the detailed reply. After reading the rebuttal, I believe the authors have clarified my concerns. I think the paper offers some valuable insights for the community.

---

> ### Author Response · Authors · 2025-08-07
> **Thanks for your reply**
>
> Thank you for your thoughtful and constructive feedback. We're happy to hear that our response clarified your concerns and that you recognize the value of our work. We will incorporate new results and analysis into our revised manuscript. Thank you again for your positive feedback!

---

### Official Review · Reviewer_M9Ae · 2025-07-01

**Clarity:** 2
**Significance:** 2
**Originality:** 2
**Rating:** 4
**Confidence:** 4

**Summary:**

This paper introduces a multi-stage approach for leveraging MLLMs for deefpake detection.

First, a frozen MLLM is asked to generate a list of questions for assessing whether some deepfake characteristics are present in an image (ie skin color consistency, mouth blurriness...). Then each of these questions is passed to a frozen along with deepfake data MLLM to observe whether the MLLMs is capable to detect the characteristics described in these questions (blurriness, consistency... the authors call them artifacts or sometimes features, which confusing with respect to actual features as in a feature space).
This assessment helps defining what questions/artifacts the MLLM is capable of detecting in a zeroshot way. The questions then are ranked and a top-k are selected to stage 2.
Stage 2 consists in using the questions of stage 1 to generate textual prompts (real and fake) to augment some deepfake dataset.
This helps building a VQA-style dataset that can be used to finetune an MLLM and making better what it already is. The intuition of finetuning is since the MLLMs is already good in zeroshot on these question, finetuning, would make it better.
Since the MLLM cannot pick up all possible artifact, the authors incorporate the predictions of a blending-based detector (the fake-it-till-you-make-it one) to compensate for the lack of performance of the MLLM for some questions/artifacts. The output of this specialized detector is also used as an annotation.
 The final model is finetuned using the VQA style built dataset which contains the initial textual prompts from stage 2 and the score of the specialized detecto

**Questions:**

Other/questions:
It would be nice if Tab2 indicated an arrow regarding the metrics to indicated whether higher/lower is better
What would be the results if the model used LAA-Net as a dedicated detector (although this still remains within the assumption that MLLMs are only performing badly on blending-based artifacts)
Why is "Photoshop traces" irrelevant aren't Adobe using AI in their software nowadays?
Why is the title including \chi as X, it sounds phonetically wrong.
Why is the work does not generalize to other types of deepfakes, doesn't this mean that the generated questions are not actually indicative enough of deepfake artifacts, or that maybe they are biased towards faceswap?

**Ethical Concerns:**

["NO or VERY MINOR ethics concerns only"]

**Final Justification:**

The authors have provided a solid rebuttal. I decided, therefore, to increase my rating. It is recommended to include all the new elements discussed in the rebuttal in  the final version of the paper.

**Limitations:**

Yes

**Paper Formatting Concerns:**

No issues.

**Quality:**

2

**Strengths And Weaknesses:**

Strengths:
- Extensive implementation details.
- Several datasets are considered, including DF40

Weaknesses:
- Unclear added scientific value/novelty, although the work introduces a sophisticated pipeline for explainable forgery detection, it remains incremental and mostly seem like an engineering solution rather than a research one
- In the related works, the limitations of existing benchmarks on forgery detection using VQA i.e. [33,19,29] are unclear.
- Several typos: Fig 1 caption: "improvsed", L38 "relies", L82 "there", L170, L173, \mathcal{M} is undefined.
- The writing style is often vague, for examples the fact of referring to the questions/artifacts learned by the model as "features", or statements like "learning to learn" etc, so, overall the writing could benefit from more clarity.
- The reliance on a blending-based detector as the dedicated detectors assumes that MLLMs are only bad in blending-based forgeries, whereas they can be bad in on other types of artifacts.
- The performance on CDF, DFD and DFDC is extremely close LAA-Net which is a simpler blending-based approach. This makes the proposed very complex in comparison to dedicated detector, which only requires generating self-blended images on the go.
- Too many acronyms making the paper hard to read
- It is unclear whether the generated question are reliable, MLLMs are can often give different answers making the annotations and mostly random. Was there some process that ensured that the questions being ranked are actually reliable, like some sort of asking the same question 100 times and average over the answers?
- The claims on extendability are not convincing and seem overstated, if by being extendable the authors mean using off-the-shelf existing approaches to generate pseudo-labels, build a dataset and train an existing architecture, then previous methods such as Face-Xray and sbi are also extendable.
- The complexity of the proposed approach is unclear.

---

> ### Author Rebuttal · Authors · 2025-07-31
>
> Dear Reviewer M9Ae，
>
> Thank you for your thoughtful comments. We hope the following responses alleviate your concerns.
>
> ---
>
> >**Q1.** Unclear added scientific value/novelty, mostly seem like an engineering solution rather than a research one.
>
> **R1:** We thank the reviewer for the feedback. We would like to clarify that our core contribution is not merely an engineering pipeline, but a **novel framework for adapting MLLMs to deepfake detection**.
>
> Specifically, our unique scientific contributions include:
>
> * **We are the first work to systematically assess the intrinsic capabilities of MLLMs for deepfake detection**, identifying its 'strong' and 'weak' features and then prioritizing features the model can reliably detect (strong features), which is new in the whole field.
> * **We are the first work that proposes an adaptive framework to effectively fuse and leverage the complementary capabilities of MLLM and SFD for comprehensive detection.** In this manner, we fully leverage the MLLM's strengths while supplementing its weaknesses by involving specific domain priors from SFDs.
>
> Overall, we believe this framework represents a significant research contribution and scientific value to the whole community.
>
> ---
>
> >**Q2.** The reliance on a blending-based detector as the dedicated detectors assumes that MLLMs are only bad in blending-based forgeries, whereas they can be bad in on other types of artifacts.
>
> **R2:** We appreciate the reviewer for this valuable point. We would like to clarify two key points:
> - **We do NOT consider that MLLMs perform poorly only in detecting blending-based forgeries**—*blending traces are merely a prominent example used for illustration*.
>     - The weaknesses of MLLMs, as identified by our MFA and supported by previous studies [1, 2], mainly lie in detecting fine-grained, low-level clues, and blending artifacts are just one notable instance of such clues. To address this limitation, our framework is designed to adaptively integrate and leverage the strengths of both MLLMs and specialized detectors (SFDs), regardless of the type of artifacts involved.
> - **Our proposed framework is NOT limited to supporting blending-based SFDs**; instead, it is designed to be *fully extendable to and compatible with any dedicated detector*.
>     - To demonstrate the flexibility and effectiveness of our framework in handling diverse artifact types, **we have integrated a new SFD specifically trained on diffusion-based artifacts**.
>     - Results in **Table 1** confirm that our framework is **not restricted to blending artifacts but is broadly effective across different types of forgeries**.
>
> **Table 1: Comprehensive performance evaluation across diverse forgery artifacts, evaluation metric is AUC**
> |Model|CDF|Simswap-DF40|Diff-FE|Diff-I2I|Diff-T2I|
> :-|:-:|:-:|:-:|:-:|:-:
> **CDFA (ECCV2025)**|87.9|76.1|74.6|81.7|87.3
> **X2DFD(Ble-CDFA)**|90.4|85.1|82.1|81.7|88.3
> **X2DFD(Ble+Diff)**|90.3|88.5|92.1|88.6|92.2
>
> ---
>
> >**Q3.** The performance on CDF, DFD and DFDC is extremely close LAA-Net which is a simpler blending-based approach. This makes the proposed method very complex compare to dedicated detector, which only requires generating self-blended images on the go.
>
> **R3** Thanks for the great question and your concern. Although the MLLM is larger than traditional detectors, the extra capacity enables it to learn richer, more diverse cues for real-vs-fake discrimination. As **Table 2** shows, when trained with our strategy the MLLM alone already outperforms the blending-based baseline, confirming that the additional parameters are effectively utilized for a more general detection, not merely overhead.
>
> Additionally, we must claim that **evaluating *only* on the three datasets CDF, DFD, and DFD, is not fair to compare** the performance between MLLM and those blending-based methods. Here is the reason:
> - **The three benchmarks are all "blending-based" deepfakes**. CDF, DFD, and DFDCP, mentioned by the reviewer, are created with face-swapping pipelines that explicitly include a final "blending" step to merge the donor face into the target image.
> - **Modern deepfakes might no longer need blending**. Recent generators such as SimSwap, InSwapper can directly synthesize the entire swapped face without any explicit blending stage.
>     - The DF40 benchmark (NeurIPS 2024) already demonstrates that blending artifacts are largely absent in these new forgeries.
>     - Therefore, performance on CDF/DFD/DFDCP alone does not reveal how well a detector generalizes to current, non-blending deepfakes.
>     - To provide a fair comparison, we re-evaluated both LAA-Net (blending-based) and our model X2DFD(Ble+Diff) on different forgery types in **Table 2**.
> - **Our MLLM-based model can learn more general patterns across different forgery types, even for those latest ones without blending:**
>     - **LAA-Net drops sharply on non-blending forgeries**, confirming its reliance on blending artifacts.
>     - **Our MLLM-based X2DFD remains stable**, showing stronger generalization beyond specific artifacts.
>
> **Table 2 Experiments comprehensive datasets compare to LAA-Net, evaluation metric is AUC.**
>
> |Model|DFDC|Simswap-DF40|Uniface-DF40|SDXL-Diff|HPS-Diff|**AVG**|
> :-|:-:|:-:|:-:|:-:|:-:|:-:
> **LAA-Net(CVPR2024)**|78.5|80.9|80.7|78.8|80.8|79.9
> **MllM only**|79.2|83.3|84.5|83.2|88.7|83.8
> **X2DFD(Ble+Diff)**|**83.2**|**88.5**|**90.1**|**89.5**|**95.4**|**89.3**
>
> ---
>
> >**Q4.** What would be the results if the model used LAA-Net as a dedicated detector (Although the model is still blending-base)
>
> **R4:** Following your suggestion, we conduct new experiments **integrating** different blending-based SFDs **LAA-Net**, and compared their performance with our original **CDFA**-based version. The results are presented in **Table 3**, which clearly indicate that our **X2DFD framework demonstrates strong and consistent performance regardless of the specific blending-based detector used**.
>
> **Table 3: Performance with different blending-based SFDs, evaluation metric is AUC**
> |Model|CDF|Uniface|Diff-FE|Diff-I2I|
> :-|:-:|:-:|:-:|:-:
> X2DFD(Ble-LAANet)|89.8|86.3|80.5|81.2
> X2DFD(Ble-CDFA)|90.4|85.5|82.1|81.7
>
> ---
>
> >**Q5.** It is unclear whether the generated questions are reliable, MLLMs can often give random answers.
>
> **R5:** Thank you for this important question on reliability. We address this with a two-part strategy to ensure our generated questions are reliable **(detailed in lines 320-336 in the manuscript)**:
>
> * **Human-in-the-Loop Filtering:** We use a hybrid process where an MLLM first generates a candidate pool of questions. Our annotators then manually **filter this pool to remove any ambiguous or low-quality questions**. This crucial step ensures only reliable questions are used for ranking.
>
> * **Deterministic Ranking:** To eliminate randomness during the ranking process itself, we use a low temperature setting (**T=0.2**). This standard practice [1,2] makes the MLLM's output highly consistent and effectively deterministic for any given input.
>
> ---
>
> >**Q6.** The claims on extendability are not convincing and seem overstated, if by being extendable the authors mean using off-the-shelf existing approaches to generate pseudo-labels, build a dataset and train an existing architecture, then previous methods such as Face-Xray and sbi are also extendable.
>
> **R6:** We sincerely thank the reviewer for this insightful comment, which allows us to further clarify the novel and distinctive nature of "extendability" in our framework.
>
> - Our notion of "extendability" is **fundamentally different from prior works** like Face-Xray or SBI, which focus on training detectors using pseudo-fakes. Instead, we leverage the MLLM as an intelligent "agent"—capable of reasoning and tool use—to **dynamically integrate diverse expert detectors (e.g., blending- or diffusion-based) as complementary sources of forensic priors**.
>     - The MLLM’s large capacity and proven ability to call and interpret tools allow it to adaptively fuse expert outputs, achieving synergistic performance (1+1 > 2). In contrast, **small models alone struggle to learn and combine such diverse patterns due to limited capacity and potential conflicts.**
> - **Prior methods are not competitors, but potential expert modules within our framework.** Our contribution is the architecture—the "how" and "who"—that enables flexible, plug-and-play integration of specialized "what" modules. This agent-expert paradigm represents a novel, extensible approach to forgery detection, distinct from traditional ensemble or training-based methods.
>
> ---
>
> >**Q7.** Why is the work does not generalize to **other types of deepfakes.**
>
> Thank you for raising this important question. Following your suggestion, we have now validated across a much broader range of deepfakes. From our results in **Table 2 of reviewer uoAb**, we can see that **our framework demonstrates strong generalization across diverse forgery types.**
>
> ---
>
> >**Q8.** The complexity of the proposed approach is unclear.
>
> **R8:** Thank you for asking about computational complexity. We measure this by calculating the average inference time per image on a single NVIDIA 4090 GPU. The results are shown in **Table 4**.
>
> **Table 4: Comparison of average inference time per image.**
> |Model|Average Inference Time (s/image)|
> :-|:-:
> **Llama-3.2-11B**|3.3
> **Qwen2.5 VL-Instruct 7B**|1.6
> **LLaVa-7B**|1.2
> **Ours (X2DFD w/ two SFDs)**|1.3
>
> ---
>
> >**Q9.** Minor Clarifications and Presentation.
>
> **R9:** Thank you very much for the detailed feedback and valuable suggestions! We will revise the manuscript to address all the points raised.
>
> ---
>
> **Reference:**
>
> [1] A learning algorithm for Boltzmann machines. Cognitive Science, 1985
>
> [2] Distilling the knowledge in a neural network. arXiv, 2015

---

> > ### Author Response · Authors · 2025-08-07
> >
> > Dear Reviewer M9Ae,
> >
> > Thank you for your valuable and constructive feedback. We have carefully reviewed your comments and made our best effort to address all of your concerns. We look forward to your response and are happy to discuss further.
> >
> > To briefly summarize, here are our specific responses to your concerns:
> > - **Clarifying Core Contributions:** We clarified the core scientific value of our work. (R1)
> > - **Effectiveness on Diverse Forgery Types:**  We added a new diffusion-based SFD and tested the framework's generalization on other types of deepfakes. (R2, R7)
> > - **Comparing with and Integrating LAA-Net:** We both compared against LAA-Net and integrated it into our framework. (R3, R4)
> > - **Reliability of Generated Questions:** We clarified how we ensure the reliability of the generated questions and reduce randomness. (R5)
> > - **Clarifying Extendability:** We clarified our unique concept of "extendability". (R6)
> > - **Complexity of the Proposed Approach:** We added the specific inference time per image. (R8)
> >
> > Thank you once again for your time and effort in reviewing our work. We are happy to continue to address any further concerns you may have.
> >
> > Sincerely,
> >
> > The Authors

---

> ### Comment · Reviewer_M9Ae · 2025-08-07
> **Rebuttal**
>
> Thanks for the valuable rebuttal that has provided several clarifications. I suggest that the authors include all these elements in the final version in case of acceptance. One concern remains: the authors do not compare their approach in terms of computational cost to standard methods (not based on MMLMs). This is extremely important to assess the practical relevance of this approach.

---

> ### Author Response · Authors · 2025-08-08
> **Thanks for your reply**
>
> We sincerely thank you for your detailed and valuable feedback on our work. We are pleased that our rebuttal has addressed most of your initial concerns, and we will incorporate all of your suggested clarifications into the revision.
>
> To address **your remaining concern about computational cost**, we provide an additional comparative analysis here. The results, summarized in **Table 1**, highlight the trade-off between inference time and performance (for both detection and explanation). Specifically, the analysis reveals three key insights:
>
> - First, MLLM-based models, with their larger parameters, are able to learn a more robust, rich, and comprehensive representation for discrimination. In contrast, non-MLLM-based smaller models (e.g., LAA-Net, CDFA), typically rely on specific forgery patterns like blending (see **R3** for details). However, the increased model size of MLLMs can lead to slower inference speeds compared to their smaller non-MLLM counterparts. Specifically,
>     - We can see from **Table 1** that MLLM baseline (w/o Explanation, using real and fake text for training) achieves notably improved detection performance compared to the non-MLLM SOTA (CDFA): **0.79 vs 0.75** on average, and largely outperforms other non-MLLM baselines.
>     - At the same time, due to the larger parameters, the MLLM-based method results in ~3X slower inference time than non-MLLM models.
>     - This indeed highlights the crucial trade-off between achieving strong detection performance and maintaining fast inference speed.
>
> - Second, providing an additional textual explanation using an MLLM requires more inference time, as the MLLM, due to its inherent characteristics, needs to generate the next token *autoregressively*. However, longer response length allows the LLM to perform more comprehensive thinking and reasoning, which can further improve the performance (as evidenced by Deepseek-R1). Specifically,
>     - Results in **Table 1** show that, when adding a longer explanation, our method not only provides **high-quality** and **intuitive explanations** for discrimination (beyond simply binary output), but also achieves significantly **stronger performance** over both MLLM and non-MLLM baselines.
>     - Similarly, adding more detailed explanations (~ 50 words) will further increase the inference time (~13X) compared to the MLLM baseline that only outputs a real/fake label without any explanation.
>
> - Third, accelerating MLLM inference is a highly active and important area of research in practice. **Our approach, based on MLLM, can directly benefit from the rapid advancements in MLLM acceleration** such as **inference frameworks** like vLLMs [1] for deployment, **post-training model compression** [2], and **model distillation** [3], which can all enhance its practical efficiency. This aligns with the typical development trajectory of most new technologies: **initial efforts focus on achieving strong performance, followed by iterative optimizations for speed, efficiency, and scalability**.
>
>
> This additional experiment highlights the crucial trade-offs between good performance, fast inference, and detailed textual explanation. Notably, achieving robust performance with rich, interpretable explanations typically requires greater computational resources and longer inference times. Inspired by your suggestion, we recognize the importance of this trade-off and plan to investigate it further in future work.
>
> Thank you once again for the discussion and your valuable suggestions! We truly appreciate your thoughtful feedback and hope that our responses have adequately addressed your concerns. We remain open to any further questions or comments and would be happy to continue the discussion.
>
>
> | Category | Model | AUC-Simswap | AUC-Diff-FE | **AUC (Average)** | Inference Time (seconds) | Textual Explanation |
> | :--- | :--- | :--- | :--- | :--- | :--- | :--- |
> | Not MLLMs | Xception | 0.68 | 0.59 | 0.64 | \~ 0.03 | None |
> | Not MLLMs | F3Net | 0.77 | 0.61 | 0.69 | \~ 0.03 | None |
> | Not MLLMs | CDFA | 0.76 | 0.74 | 0.75 | \~ 0.05 | None |
> | MLLMs | w/o Explanation | 0.79 | 0.78 | 0.79 | \~ 0.09 | None |
> | MLLMs | Ours | 0.89 | 0.92 | 0.91 | \~ 1.3 | Good |
> ---
> **Reference:**
>
> [1] Efficient Memory Management for Large Language Model Serving with PagedAttention
>
> [2] LLaVolta: Efficient Multi-modal Models via Stage-wise Visual Context Compression
>
> [3] LLaVA-KD: A Framework of Distilling Multimodal Large Language Models

---

> > ### Comment · Reviewer_M9Ae · 2025-08-08
> > **Increasing my rating**
> >
> > Thanks a lot for those additional elements. I suggest including the discussion about computational cost and the pros/cons of MLLMs in the paper as compared to more traditional methods.  This is very useful for the community and brings an objective point of view regarding the current state-of-the-art. I increase my rating.

---

> > > ### Author Response · Authors · 2025-08-08
> > >
> > > We very much thank you for your detailed and insightful feedback! You are absolutely right that including a comparative analysis of computational costs, along with a thoughtful discussion of the advantages and limitations of MLLMs, is essential. Such an addition would largely strengthen our manuscript and contribute meaningfully to the broader research community.
> > >
> > > Thank you again for lending your expertise and helping us make our paper so much better.

---

### Official Review · Reviewer_uoAb · 2025-07-02

**Clarity:** 4
**Significance:** 4
**Originality:** 3
**Rating:** 5
**Confidence:** 5

**Summary:**

This paper presents a detailed exploration of the potential of MLLM for deepfake detection. It focuses on one of the main challenges in this area — how to build a good image-text dataset. The author raise two central questions — whether MLLM can interpret forgery clue descriptions generated by humans or GPT, and which types of clues are more effective — and provide preliminary experimental validation. They further propose enhancing the model's strong features while supplementing weak ones using external deepfake detectors. A concrete framework is introduced and validated through extensive experiments.

**Questions:**

1. Why is fine-tuning MLLM only with strong features more effective than using all available features? The paper demonstrates this via experiments, but there’s not much explanation behind it.
2. As noted, the proposed method still heavily relies on the capabilities of specific detectors. Could the authors analyze the deeper reason behind this? Is it because the visual encoder of the MLLM fails to capture fine-grained forgery cues?
3. For deepfake images/videos without obvious artifacts, how can one obtain accurate and reliable textual descriptions to effectively utilize the capabilities of MLLM?

**Ethical Concerns:**

["NO or VERY MINOR ethics concerns only"]

**Final Justification:**

The authors have patiently and thoroughly addressed the concerns I raised, providing clear explanations and additional clarifications where necessary. While I appreciate their efforts and responsiveness, my overall assessment of the paper remains unchanged, and I am therefore maintaining my original rating.

**Limitations:**

yes

**Quality:**

4

**Strengths And Weaknesses:**

### Strengths
1. The paper suggests a good way to generate text descriptions of forgery features with little manual effort, which helps solve a key problem when using MLLM for deepfake detection.
2. It presents a well-designed integration framework that leverages the complementary strengths of MLLM and Specific Feature Detector.
3. The paper conducts thorough ablation studies to demonstrate the significance and effectiveness of feature selection and other individual components.
4. The paper is well written, easy to follow, and the ideas are clearly explained.

### Weaknesses
1. The overall performance boost comes from using the SFD module. It’s not very clear if MLLM alone can really beat traditional deepfake detection models.
2. The method works well on datasets with clear fake artifacts, but it may not be strong enough for high-quality fakes, like those made by diffusion models. This is understandable, but still a limitation.

---

> ### Author Rebuttal · Authors · 2025-07-31
>
> Dear Reviewer uoAb，
>
> We sincerely appreciate your time and effort in reviewing our work. We are extremely encouraged by your positive assessment and your recognition of our work's key strengths, including its **efficient data generation method, a well-designed integration framework, detailed ablation studies, and clear writing**. In the following, we would like to respond to your valuable comments separately.
>
> ---
>
> >**Q1.** The overall performance boost comes from using the SFD module. It’s not very clear if MLLM alone can really beat traditional deepfake detection models.
>
> **R1:** Thank you for your great observation. To verify the contribution with MLLM alone, we conduct an additional ablation (see **Table 1**), comparing the generalization performance among SFD, MLLM alone, MLLM + SFD (ours).
>
> We summarize the key observations below:
>
> - **The MLLM alone is also competitive and can beat traditional models on certain testing datasets.** As shown in the "Average" column, MLLM only, trained using our proposed strategy, achieves an average AUC of **82.6%**, demonstrating competitive—and even superior—average performance compared to the SOTA traditional detector CDFA **(81.0%)**, a SFD specifically designed for capturing blending artifacts.
> - **SFDs typically perform well on specific types of fakes but fail to maintain generalization across all types.** As shown in **Table 1**, CDFA (a blending-based SFD) performs very well on blending-based deepfakes such as CDF and DFDC (where additional blending operations are used in face-swapping, leaving strong blending artifacts), but struggles on non-blending face-swapping methods (where entire images are generated), achieving only 76.1% and 76.5% on SimSwap and UniFace, respectively. **In contrast, *MLLM only* surpasses CDFA by nearly 10% on these cases.**
>     - This finding validates our core finding: **the MLLM is a strong generalist detector, yet it exhibits specific low-level "weaknesses"** (not as effective as SFDs specifically designed to capture certain fine-grained artifacts). This is precisely why our "Weak Feature Supplementing" (WFS) module is necessary, and why the full X2DFD framework achieves the best overall performance.
>
>
> **Table 1: Ablation study comparing the performance of SFD, MLLM only, and ours (measured by AUC\%)**
> | | CDF | DFDC | Simswap | Uniface | **Average** |
> | :--- | :--: | :--: | :--: | :--: | :---: |
> | **SFD (CDFA, ECCV 2024)** | 87.9 | 83.5 | 76.1 | 76.5 | 81.0 |
> | **MLLM only** | 83.2 | 79.2 | 83.3 | 84.5 | 82.6 |
> | **SFD + MLLM (ours)** | 90.4 | 83.7 | 85.1 | 85.5 | **86.2** |
>
>
>
>
> ---
>
> >**Q2.** As noted, the proposed method still heavily relies on the capabilities of specific detectors. Could the authors analyze the deeper reason behind this? Is it because the visual encoder of the MLLM fails to capture fine-grained forgery cues?
>
>
>
>
> **R2:** Thank you for the question. We would like to clarify that our proposed framework does **not heavily rely on specific feature detectors (SFDs) in a dependent or restrictive manner**. Instead, it is designed as an adaptive fusion approach that **leverages two complementary components**: the MLLM and the SFD.
>
> - Specifically, the MLLM excels at identifying general, high-level inconsistencies and semantic anomalies in images, while the SFD is specialized in detecting fine-grained, localized artifacts—such as blending boundaries or texture irregularities—that are often indicative of image manipulation. **By adaptively combining these two complementary signals, our framework achieves a synergistic effect**, where the whole is greater than the sum of its parts ("1+1 > 2") in terms of detection performance.
> - This design choice is **grounded in evidence from recent studies**, such as [1,2], which demonstrate that MLLMs and traditional detectors capture different types of forgery cues and thus offer complementary strengths. Therefore, rather than depending solely on SFDs, our method strategically integrates their fine-grained sensitivity with the broader contextual understanding of MLLMs, leading to more robust and comprehensive detection.
>
>
>
> ---
>
> >**Q3.** The method works well on datasets with clear fake artifacts, but it may not be strong enough for high-quality fakes, like those made by **diffusion models**. This is understandable, but still a limitation.
>
>
> **R3:** Thanks for your very insightful comment! Following your suggestion, we add an additional experiment for testing our model on high-quality diffusion-based fakes, suggested by the reviewer. Specifically:
> - We test our models on the **DiFF dataset** [3], a **diffusion-based** face forgery dataset. We then incorporate a **new SFD specialized in detecting diffusion artifacts** [4].
>
> - **Specifically, our framework maintains effective performance on diffusion-based forgeries.** Results in **Table 2** demonstrate that the MLLM trained by our pipeline achieves **significant improvement** over the traditional baseline (Xception),
>     - Furthermore, we see that **adding an additional SFD (a diffusion-based expert) can further enhance** the detection performance. This highlights the advantage of our framework's extendability, as integrating a domain-specific priors like SFD yields substantial improvements.
>
> - Additionally, it is worth noting that **MLLM alone can also learn some *generalizable* discriminative features for detection.**  Although not all these features are explainable for human.
>     - Results in **Table 3** show that human performance (leveraging human-understandable features for detection) is substantially lower than that of the MLLM. This indicates the MLLM can capture certain discriminative patterns beyond human perception—features that are generalizable even if not fully explainable to humans.
>
>
> **Table 2 Practical extension and validation for diffusion-based forgery, evaluation metric is AUC**
>
> | | CDF | Simswap | Diff-FE | Diff-FS | Diff-I2I | Diff-T2I |
> | :--- | :---: | :---: | :---: | :---: | :---: | :---: |
> | **Xception** |73.7 |77.6 | 58.6 | 85.9 | 56.8 | 62.4 |
> | **MLLM alone** | 83.2 | 83.3 | 80.1 | 90.1 | 80.3 | 84.2 |
> | **X2DFD(w/ SFD)** | **90.3** |**88.5** |**92.1** |**97.2** |**88.6** | **92.2**|
>
>
>
> **Table 3 Experiments on without obvious artifacts, evaluation metric is fake ACC**
>
> | | FreeDoM_T | CoDiff |
> | :--- | :---: | :---: |
> | **Human performance (ACC)** | 25.4 | 27.8 |
> | **MLLM only** | 59.8 | 53.4 |
> | **X2DFD (W/ SFD)** | **79.9** | **70.5** |
>
> ---
>
> >**Q4.** For deepfake images/videos **without obvious artifacts**, how can one obtain accurate and **reliable textual descriptions** to effectively utilize the capabilities of MLLM?
>
>
>
> **R4:** Thank you very much for this meaningful quesiton. However, please note that obtaining a completely accurate and reliable caption for realistic deepfakes without obvious artifacts is highly challenging and remains an open problem for the entire field.
>
> Generally, we think there are two main solutions to measure the reliability of MLLM's explanations:
> - For those **semantically understandable features**, **human** can act as a reasonable judger (see the **human study of our manuscript**), since human (experts) are capable of understanding most high-level semantic concepts.
> - For those **low-level detection signals**, it is very difficult for humans to interpret (but might be captured by machine). This is because **not all discriminative features are human-explainable**. In such cases, we argue that an **SFD (as a domain expert) can offer a more reliable prediction than a human**, especially when its prediction is made with high confidence. This is precisely why we design an extendable framework that allows integration of any SFD to enhance the overall reliability of the final decision.
>
> In our framework's implementation, guided by these two principles, we first instill human semantic knowledge into the MLLM during training, enabling it to generate human-understandable explanations. Then, for more realistic fakes where low-level artifacts dominate, we feed the SFD's prediction into the MLLM, allowing it to produce a **final explanation that includes both the human's understanding and SFD's domain-specific prior**, thereby improving overall reliability.
>
>
> To summarize, evaluating explanation reliability—especially for highly realistic fakes—remains an open challenge. When features are semantically interpretable, humans can act as judges; otherwise, we should rely on domain experts (i.e., SFDs). This dichotomy **directly motivates the design of our extendable framework.**
>
> ---
>
>
>
> >**Q5.** Why is fine-tuning MLLM only with **strong features more effective than using all available features**? The paper demonstrates this via experiments, but there’s not much explanation behind it.
>
> **R5:** Thank you for this question. Fine-tuning on "strong features" is more effective because it focuses the learning process on **a set of highly discriminative features**, whereas including "weak features" introduces disruptive noise. Specifically,
>
> * **Focusing on highly discriminative features**: Fine-tuning only on "strong features" concentrates the learning process on **reliable, relatively high-quality forgery features** that the MLLM "familiar with" and can effectively utilize for detection.
> * **Avoiding disruptive noise**: Including weak features in the training can introduce disruptive noise. Our preliminary investigation has demonstrated that the MLLM struggles to use these weak features for detection (resulting in lower detection accuracy).
>
>
> ---
>
> **References:**
>
> [1] Q-bench: A benchmark for general-purpose foundation models on low-level vision. ICLR, 2024
>
> [2] Can chatgpt detect deepfakes? a study of using multimodal large language models for media forensics. CVPR Workshop, 2024
>
> [3] Diffusion Facial Forgery Detection. ACMMM, 2024
>
> [4] Aligned Datasets Improve Detection of Latent Diffusion-Generated Images. ICLR, 2025

---

> > ### Comment · Reviewer_uoAb · 2025-08-07
> >
> > Thank you for the authors’ response, which has largely addressed my concerns. This work provides valuable insights and effective results in applying MLLMs to deepfake detection. I will maintain my rating.

---

> > > ### Author Response · Authors · 2025-08-07
> > > **Thanks for your reply**
> > >
> > > Thank you very much for your time and constructive feedback. We truly appreciate your thoughtful review and your recognition and recommendation for our work. We will incorporate new results and analysis into our revised manuscript. Thank you once again!

---

### Note · Authors · 2025-08-12

Dear Area Chair,

We sincerely thank all reviewers for their time, constructive feedback, and the recognition of our work:
* **Novelty and Insightful Idea** (Reviewers uoAb, 6sJc, 1fa1)
* **Comprehensive and Effective Experimental Results** (Reviewers uoAb, M9Ae, 6sJc)
* **Clarity, Presentation, and Extensive Implementation Details** (Reviewers uoAb, 1fa1, M9Ae)

Our work, X²-DFD, introduces an explainable and extendable framework. It first assesses a Multimodal Large-Language Model's (MLLM's) strengths and weaknesses, then enhances its strengths by prioritizing strong features for explanations, and finally supplements its weaknesses using specific feature detectors (SFDs).

During the rebuttal period, the reviewers raised many valuable suggestions. We conducted active interactions and deep discussions, and supplemented our work with detailed experiments to clarify their concerns. After the discussion period, we are happy to see that **all reviewers' concerns have been addressed**.

We hope this work, which introduces an explainable and extendable framework built on the principle of 'enhancing strengths and supplementing weaknesses', advances MLLM-based deepfake detection and contributes to the broader research community.



Thank you once again for your time and careful consideration.

Sincerely,

The Authors of Submission 14722

---

### Decision · Program_Chairs · 2025-09-17

**Decision:**

Accept (poster)

**Comment:**

This paper proposes an explainable and extendable framework for deepfake detection leveraging MLLMs. Reviewers found the idea novel, the framework well-designed, and experiments comprehensive, with strengths in modularity and clarity. However, concerns were raised about reliance on specific detectors, incremental novelty, complexity, and generalization to high-quality fakes. The rebuttal addressed most concerns with additional experiments and clarifications, improving confidence in the contributions. Overall, the submission presents a promising and timely direction with strong empirical support, though some limitations remain.